# Promoter sequence and architecture determine expression variability and confer robustness to genetic variants

Hjörleifur Einarsson, Marco Salvatore, Christian Vaagensø, Nicolas Alcaraz[†], Jette Bornholdt[‡], Sarah Rennie, Robin Andersson*

Department of Biology, University of Copenhagen, Copenhagen, Denmark

**Abstract** Genetic and environmental exposures cause variability in gene expression. Although most genes are affected in a population, their effect sizes vary greatly, indicating the existence of regulatory mechanisms that could amplify or attenuate expression variability. Here, we investigate the relationship between the sequence and transcription start site architectures of promoters and their expression variability across human individuals. We find that expression variability can be largely explained by a promoter's DNA sequence and its binding sites for specific transcription factors. We show that promoter expression variability reflects the biological process of a gene, demonstrating a selective trade-off between stability for metabolic genes and plasticity for responsive genes and those involved in signaling. Promoters with a rigid transcription start site architecture are more prone to have variable expression and to be associated with genetic variants with large effect sizes, while a flexible usage of transcription start sites within a promoter attenuates expression variability and limits genotypic effects. Our work provides insights into the variable nature of responsive genes and reveals a novel mechanism for supplying transcriptional and mutational robustness to essential genes through multiple transcription start site regions within a promoter.

**\*For correspondence:**
robin@bio.ku.dk

**Present address:** [†]Novo Nordisk Foundation Center for Protein Research (CPR), University of Copenhagen, Copenhagen, Denmark; [‡]Adcendo ApS, Copenhagen, Denmark

**Competing interest:** The authors declare that no competing interests exist.

## Editor's evaluation

This paper presents valuable findings about how human genetic variation impacts gene expression. Using a compelling analysis of new experimental data based on cell lines from 108 individuals, the authors uncover features that distinguish promoters with highly variable expression across individuals from those exhibiting low variability. This work and the associated resource will be of broad interest for further investigations of the interplay between genetic variation and gene expression control.

## Introduction

Transcriptional regulation is the main process controlling how genome-encoded information is translated into phenotypes. Hence, understanding how transcriptional regulation influences gene expression variability is of fundamental importance to understand how organisms are capable of generating proper phenotypes in the face of stochastic, environmental, and genetic variation. Through differentiation, cells acquire highly specialized functions, but need to still maintain their general abilities to accurately regulate both essential pathways as well as responses to changes in the environment. To achieve robustness, regulatory processes must be capable of attenuating expression variability of essential genes (*Bartha et al., 2018*), while still allowing, or possibly amplifying (*Eldar and Elowitz, 2010*; *Urban and Johnston, 2018*), variability in expression for genes that are required for

differentiation or responses to environmental changes and external cues. How cells can achieve such precision and robustness remains elusive.

Genetic variation affects the expression level (*Montgomery et al., 2010*; *Pickrell et al., 2010*; *Stranger et al., 2007*) of the majority of human genes (*Battle et al., 2017*; *Lappalainen et al., 2013*; *Storey et al., 2007*). However, genes are associated with highly different effect sizes, with ubiquitously expressed or essential genes frequently being less affected (*Battle et al., 2017*). This indicates that genes associated with different regulatory programs are connected with different mechanisms or effects of mutational robustness (*Payne and Wagner, 2015*). Multiple transcription factor (TF) binding sites may act to buffer the effects of mutations in promoters (*Spivakov et al., 2012*), and promoters can have highly flexible transcription start site (TSS) architectures (*Akalin et al., 2009*; *Carninci et al., 2006*; *Lehner, 2008*). This demonstrates that the sequence and architecture of a promoter may influence its variability in expression across individuals.

Previous studies aimed at identifying processes involved in the regulation of gene expression variability have indeed revealed regulatory features mostly associated with the promoters of genes, such as CpG islands and TATA-boxes (*Morgan and Marioni, 2018*; *Ravarani et al., 2016*; *Sigalova et al., 2020*), the chromatin state around gene TSSs (*Faure et al., 2017*), and the propensity of RNA polymerase II to pause downstream of the TSS (*Boettiger and Levine, 2009*). These studies have relied on model organisms or focused on transcriptional noise across single cells. As of yet, regulatory features have not been thoroughly studied from the perspective of variability in promoter activity or across human individuals. Furthermore, it is unclear if regulation of variability mainly acts to attenuate variability to achieve stable expression for certain genes or if independent regulatory processes act in parallel to amplify variability for other genes.

Here, we provide a comprehensive characterization of the sequences, TSS architectures, and regulatory processes determining variability of promoter activity across human lymphoblastoid cell lines (LCLs). We find that variability in promoter activity is to a large degree reflected by the promoter sequence, notwithstanding possible genotypic differences. Furthermore, the presence of binding sites for specific TFs, including those of the ETS family, are highly predictive of low promoter variability independently of their impact on promoter expression levels. In addition, we demonstrate that differences in the variability of promoters reflect their involvement in distinct biological processes, indicating a selective trade-off between stability and plasticity of promoters. Finally, we show that flexibility in TSS usage is associated with attenuated promoter variability. Our results reveal a novel mechanism that confers mutational robustness to gene promoters via switches between proximal core promoters. This study provides fundamental insights into transcriptional regulation, indicating shared mechanisms that can buffer stochastic, environmental, and genetic variation and how these affect the responsiveness and cell-type restricted activity of genes.

## Results
### TSS profiling reveals variability in promoter activity across individuals

To characterize human variability in promoter activities, we profiled TSSs using CAGE (Cap Analysis of Gene Expression *Takahashi et al., 2012*; *Figure 1A*) across 108 Epstein-Barr virus (EBV)-transformed LCLs (*Auton et al., 2015*) of African origin, 89 from Yoruba in Ibadan, Nigeria (YRI) and 19 from Luhya in Webuye, Kenya (LWK) (*Supplementary file 1*). The samples had a balanced sex ratio, 56 females and 52 males, and no observable population stratification in the expression data (*Figure 1—figure supplement 1*). With CAGE, TSSs can be mapped with single base pair resolution and the relative number of sequencing reads supporting each TSS gives simultaneously an accurate estimate of the abundance of its associated RNA (*Kawaji et al., 2014*). The CAGE data across the LCL panel therefore give us a unique opportunity to both estimate variability in promoter activity and characterize the regulatory features influencing such variability.

We identified 29,001 active promoters of 15,994 annotated genes (*Frankish et al., 2019*) through positional clustering of proximal CAGE-inferred TSSs on the same strand (*Figure 1A*; *Carninci et al., 2006*) detected in at least 10% of individuals (*Supplementary file 2*). This individual-agnostic strategy ensured a focus on promoters that are active across multiple individuals while also allowing for the measurement of variability in promoter activity across the panel. For example, the CAGE data revealed that the promoters of gene *RPL26L1*, encoding a putative component of the large 60 S subunit of the

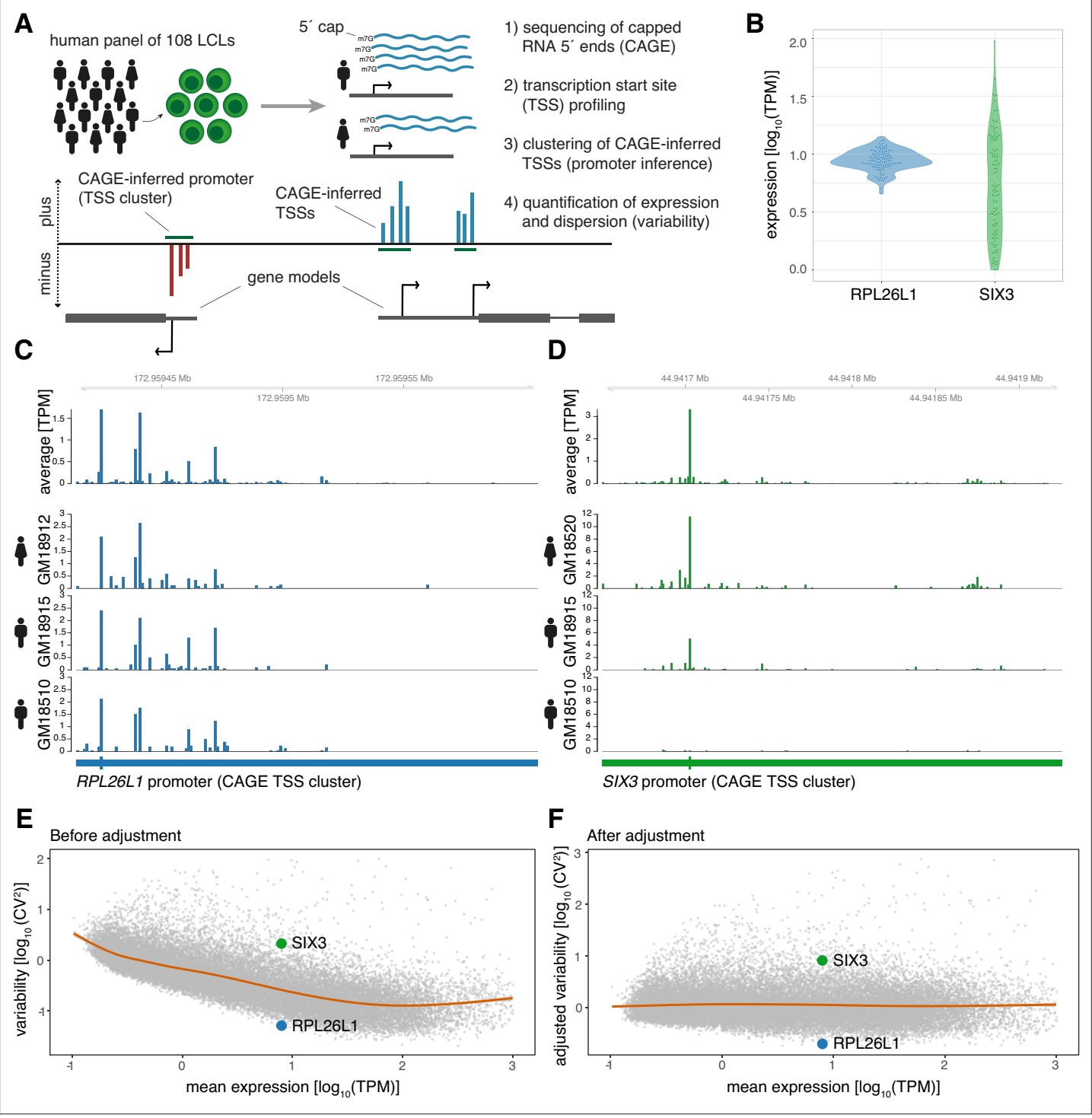

**Figure 1.** CAGE profiling of TSSs reveals diverse promoter variability across individuals. (**A**) Illustration of the experimental design and approach for measuring promoter activity and variability. Capped 5′ ends of RNAs from LCLs derived from 108 individuals were sequenced with CAGE, followed by individual-agnostic positional clustering of proximal CAGE-inferred TSSs (first 5′ end bp of CAGE reads). The expression level of the resulting CAGE-inferred promoters proximal to annotated gene TSSs were quantified in each individual and used to measure promoter variability. (**B**) Example of promoter activity (TPM normalized count of CAGE reads) across individuals for a low variable promoter (gene *RPL26L1*) and a highly variable promoter (gene *SIX3*) with similar average expression across the panel. (**C–D**) Genome tracks for two promoters showing average TPM-normalized CAGE data (expression of CAGE-inferred TSSs) across individuals (top track) and TPM-normalized CAGE data for three individuals (bottom tracks) for a low variable promoter (panel C, gene *RPL26L1*) and a highly variable promoter (panel D, gene *SIX3*). (**E–F**) The $CV^2$ (squared coefficient of variation) and mean expression relationship of 29,001 CAGE-inferred promoters across 108 individuals before (**E**) and after (**F**) adjustment of the mean expression-dispersion

*Figure 1 continued on next page*

*Figure 1 continued*

relationship. The CV$^2$ and mean expression are log$_{10}$ transformed, orange lines show loess regression lines fitting the dispersion to the mean expression level, and example gene promoters from B-D are highlighted in colors.

The online version of this article includes the following figure supplement(s) for figure 1:

**Figure supplement 1.** PCA plot of promoter expression (CAGE) across the LCL panel.

ribosome, and transcription factor gene *SIX3* have highly different variance yet similar mean expression across individuals (*Figure 1B–D*).

We used the squared coefficient of variation (CV$^2$) as a measure of promoter expression dispersion, revealing how the normalized expression across individuals deviates from the mean for each identified promoter. We observed that the promoter CV$^2$ decreases by increasing mean expression (*Figure 1E*; *Eling et al., 2018*; *Kolodziejczyk et al., 2015*; *Sigalova et al., 2020*). To account for this bias, we subtracted the expected dispersion for each promoter according to its expression level (*Kolodziejczyk et al., 2015*; *Newman et al., 2006*). Importantly, rank differences in promoter dispersion were maintained for each expression level after adjustment, as seen for promoters of genes *RPL26L1* and *SIX3* (*Figure 1E and F*). This strategy thus allowed us to investigate how promoter architecture and sequence determine variability in promoter activity across the panel separately from its impact on expression level (*Figure 1F*).

## Promoter expression variability is reflected by the promoter sequence

To investigate if local sequence features at promoters reflect their variability in activity, we applied machine learning (convolutional neural network (CNN); *Figure 2—figure supplement 1A*; see Materials and methods) to discern low variable promoters (N=5,054) from highly variable promoters (N=5,683) based on their local DNA sequence alone. We considered the genomic reference sequence to model the intrinsic component of variability encoded within the promoter sequence independently of local genetic variants within the panel. The resulting model was capable of distinguishing between these promoter classes with high accuracy (area under receiving operating curve (AUC)=0.84 for the out of sample test set; *Figure 2—figure supplement 1B*), equally well for highl and low variable promoters (per-class test set F1 scores of 0.76 and 0.77, respectively).

To assess which sequence features the CNN had learned to distinguish the classes, we calculated importance scores using DeepLift (*Shrikumar et al., 2019*) for each nucleotide in the input sequences for predicting low and high promoter variability. This approach relies on backpropagation of the contributions of all neurons in the CNN to the input features, nucleotides, and can therefore be used to identify properties or short stretches of DNA indicative of amplifying or attenuating expression variability. We applied motif discovery on clustered stretches, so called metaclusters, of the input sequences with high importance scores (*Shrikumar, 2020*) and matched the identified metaclusters to known TF binding motifs (*Fornes et al., 2020*). This strategy revealed TFs indicative of either high or low promoter variability (*Figure 2A–C*). Noteworthy, we observed motifs for the ETS superfamily of TFs, including ELK1, ETV6, and ELK3, associated with low variable promoters, and motifs for PTF1A, ASCL2, and FOS-JUN heterodimer (AP-1) among highly variable promoters. These results demonstrate that the promoter sequence and its putative TF binding sites are predictive of the expression variability of a promoter.

## Sequence features of promoters are highly predictive of promoter variability

To systematically test how predictive TF binding sites are of the variability of active promoters, we made use of binding sites predicted from motif scanning for 746 TFs (*Fornes et al., 2020*). TF binding site profiles and low/high CpG content (*Figure 2—figure supplement 2A*) were collected for each identified promoter and the resulting feature data were used to train a machine learning (random forest) classifier features associated with either high or low variability (low variable N=5054, highly variable N=5683). Feature selection (*Kursa and Rudnicki, 2010*) identified 124 of the 746 TFs as well as CpG ratio to be important for classification, and a classifier based on these selected features demonstrated high predictive performance (AUC = 0.79; per-class F1 score of 0.73 and 0.68 for low

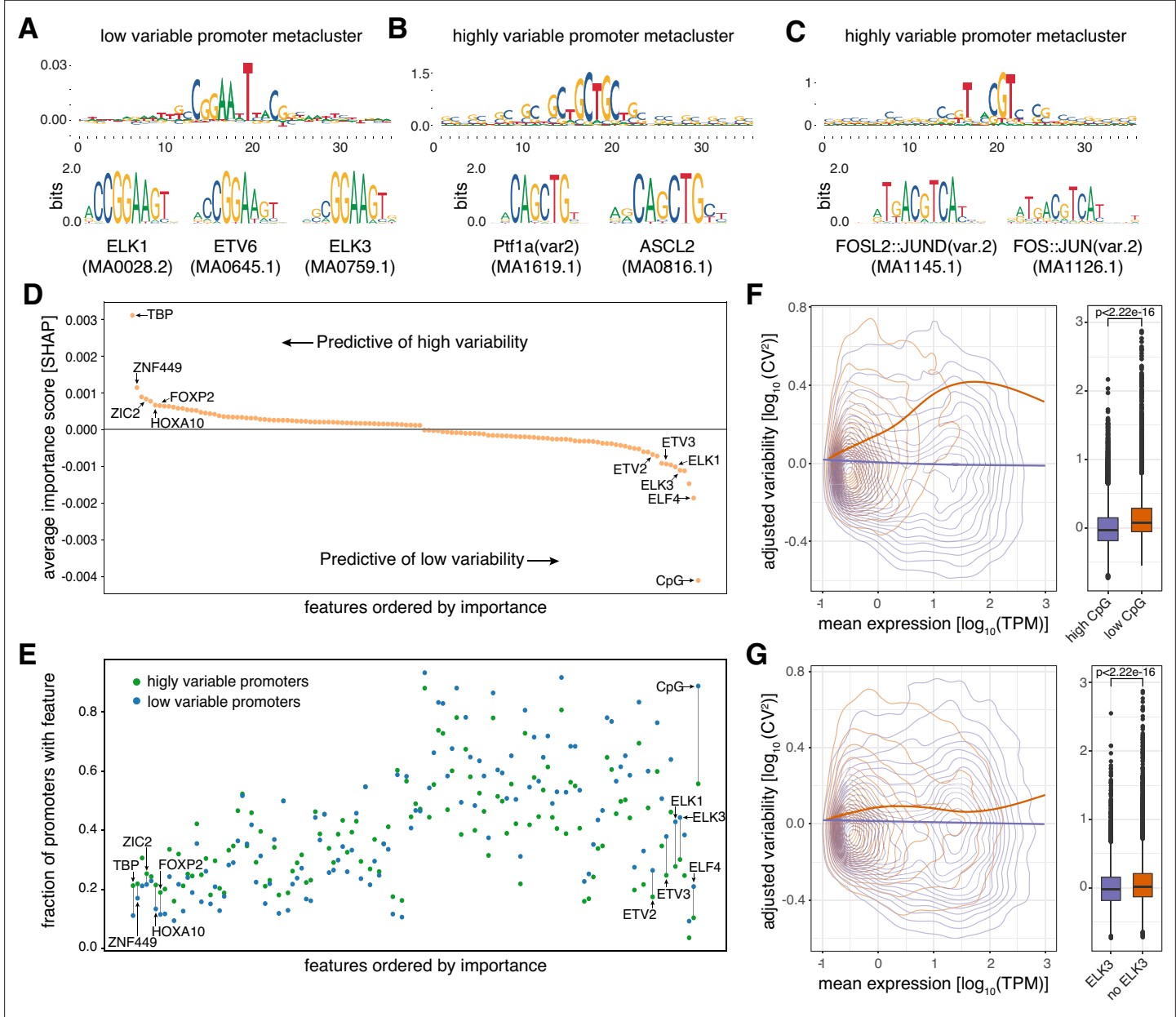

**Figure 2.** Promoter sequence features are highly predictive of promoter variability. (**A**) Sequence logo of a metacluster (top) identified for low variable promoter sequences that matches known TF motifs (bottom) for ETS factors ELK1, ETV6, and ELK3. (**B–C**) Sequence logos of two metaclusters (top) identified for highly variable promoter sequences that match known TF motifs (bottom) for PTF1A and ASCL2 (**B**) and FOSL2-JUND and FOS-JUN heterodimers (**C**). (**D**) Average contribution (SHAP values) of CpG content and each of the 124 TFs identified as important for predicting promoter variability. Features are ordered by their average contribution to the prediction of highly variable promoters and selected TFs are highlighted. For a full version of the plot see *Figure 2—figure supplement 3A*. (**E**) The frequency of predicted TF binding sites (presence/absence) in highly variable (green) and low variable (blue) promoters. TFs are ordered as in D. For a full version of the plot see *Figure 2—figure supplement 3B and C*. (**F–G**) Promoters split into groups based on the presence/absence of high CpG content (**F**), and predicted binding sites of ELK3 (**G**). For both features displayed in panels F and G, the left subpanels display the relationship between $\log_{10}$-transformed mean expression levels and adjusted $\log_{10}$-transformed $CV^2$ with loess regression lines shown separately for each promoter group. The right subpanels display box-and-whisker plots of the differences in adjusted $\log_{10}$-transformed $CV^2$ between the two promoter groups (central band: median; boundaries: first and third quartiles; whiskers:+/-1.5 IQR). p-values were determined using the Wilcoxon rank-sum test.

The online version of this article includes the following figure supplement(s) for figure 2:

**Figure supplement 1.** Neural network model and performance.

**Figure supplement 2.** Random forest features and performance.

*Figure 2 continued on next page*

*Figure 2 continued*

**Figure supplement 3.** Full panel of features found to be predictive of promoter variability.

**Figure supplement 4.** Association between TF binding sites and promoter expression level.

**Figure supplement 5.** Association between the number of ETS binding sites and promoter variability.

and highly variable promoters, respectively; *Figure 2—figure supplement 2B*), reinforcing the strong link observed between DNA sequence and promoter variability (*Figure 2—figure supplement 1B*).

Reverse engineering of the random forest classifier (SHAP, Shapley additive explanations) (*Lundberg and Lee, 2017*) allowed us to calculate the marginal contribution of each of the 125 selected features to the prediction of variability class for each promoter and whether the feature on average is indicative of amplifying or attenuating variability of expression when present in the promoter sequence (*Figure 2D*; *Figure 2—figure supplement 3A*). We identified the presence of high observed/expected CpG ratio and TATA-binding protein (TBP) binding sites (TATA-boxes) to be the strongest predictive features of low and high promoter variability, respectively. Although the remaining TFs contribute only marginally on their own to the predictions compared to TATA-box and high CpG content status (*Figure 2D*; *Figure 2—figure supplement 3A*), a baseline model (decision tree) based on CpG ratio and TBP binding site presence alone yielded worse performance than the full model (AUC = 0.71 vs 0.79 for the baseline model and the full model, respectively; *Figure 2—figure supplement 2B*). This demonstrates that the TF binding grammar contributes to a promoter's expression variability.

TFs associated with highly variable promoters are mostly related to tissue-specific or developmental regulation (e.g., FOXP2, HOXA10) while TFs predictive of low promoter variability are generally associated with ubiquitous activity across cell types and a diverse range of basic cellular processes (e.g. ELK1, ELF4, ETV3). In addition, TFs predictive of high variability (e.g. ZIC2, ZNF449, HOXA10) tend to have binding sites in relatively few highly variable promoters while TFs predictive of low promoter variability (e.g. ELK1, ELK3) show a propensity for having binding sites present in a large number of promoters (*Figure 2E*; *Figure 2—figure supplement 3B and C*). This suggests that variably expressed promoters have diverse TF binding profiles and that the regulatory grammar for promoter stability is less complex.

Although the adjusted dispersion of promoters was separated from their expression level (*Figure 1E*), we observed that the presence of binding sites for some TFs that are predictive of promoter variability are also associated with promoter expression level (*Figure 2—figure supplement 4*). Importantly, despite this association, the effect of identified features on promoter variability is still evident across a range of promoter expression levels (*Figure 2F and G*). This is particularly apparent for CpG islands, which seem to have an attenuating effect on promoter variability regardless of expression level (*Figure 2F*).

Many of the TFs identified as being predictive of low variability (e.g. ELK1, ELK3, ELF4, ETV2, ETV3) belong to the ETS family of TFs (*Figure 2D*; *Figure 2—figure supplement 3A*), a large group of TFs that are conserved across Metazoa and characterized by their shared ETS domain that binds 5'-GGA(A/T)–3' DNA sequences (*Sharrocks, 2001*). ETS factors are important regulators of promoter activities in lymphoid cells (*Hepkema et al., 2020*), but are generally involved in a wide range of crucial cellular processes such as growth, proliferation, apoptosis, and cellular homeostasis (*Kar and Gutierrez-Hartmann, 2013*; *Oikawa and Yamada, 2003*; *Suico et al., 2017*). Furthermore, different ETS factors can bind in a redundant manner to the same promoters of housekeeping genes (*Hollenhorst et al., 2011*; *Hollenhorst et al., 2007*). However, the shared DNA-binding domain of ETS factors makes it hard to discern individual factors based on their binding motifs alone (*Figure 2A*). Although in general linked with higher promoter activity (*Curina et al., 2017*), ETS binding site presence is associated with lower variability across all expression levels (*Figure 2G*). In addition, the degree of promoter variability decreases by an increasing number of non-overlapping ETS binding sites (*Figure 2—figure supplement 5A*), regardless of promoter expression level (*Figure 2—figure supplement 5B*), suggesting that multiple ETS binding sites can either facilitate cooperativity between ETS factors or provide robustness to stabilize promoter variability across individuals.

Taken together, our results indicate that promoter sequence can influence both low and high promoter variability across human individuals independently from its impact on expression level. Our results further indicate that variable promoters exhibit highly diverse binding grammars for TFs that

are associated with relatively few promoters, while a more uniform regulatory grammar is indicated for stable promoters, being highly associated with higher CpG content and ETS binding sites.

## Variability in promoter activity reflects plasticity and robustness for distinct biological functions

The high performance of predicting promoter variability from local DNA sequence and the distinct TF binding profiles of low and highly variable promoters imply that attenuation and amplification of variability are driven by distinct regulatory mechanisms. This argues that favoring robustness (low variability) over plasticity (high variability) should reflect the biological processes where this provides regulatory advantages. Supporting this hypothesis, we observed that low variable promoters were highly enriched with basic cellular housekeeping processes, in particular metabolic processes (*Figure 3A*). In contrast, highly variable promoters were enriched with more dynamic biological functions, including signaling, response to stimulus, and developmental processes.

Interestingly, the same features found to be predictive of low and high promoter variability across individuals, including CpG-content and TATA-boxes (TBP binding sites), are also associated with low and high transcriptional noise across individual cells (*Faure et al., 2017*; *Morgan and Marioni, 2018*). The presence of a TATA-box is also associated with high gene expression variability in flies (*Sigalova et al., 2020*). This suggests that some of the same underlying regulatory mechanisms that dictate low or high transcriptional noise across single cells are maintained and conserved between humans and flies at an individual level and manifested to control low and high expression variability across a population, respectively, as well as housekeeping or restricted activity across cell types.

In agreement, genes of highly variable promoters tend to have higher transcriptional noise than those of low variable ones across GM12878 single cells (Cohen's d=0.411, $p<2.2 \times 10^{-16}$, two sample t-test; *Figure 3—figure supplement 1A*; *Supplementary file 3*). Furthermore, we observed an inverse correlation between variability in promoter activity and the number of cell types (*FANTOM Consortium and the RIKEN PMI and CLST (DGT) et al., 2014*) and tissues (*Battle et al., 2017*) the corresponding gene is expressed in (Spearman's rank correlation $\rho = -0.21$ and $-0.15$ for cell types and tissues, respectively, $p < 2.2 \times 10^{-16}$; *Figure 3B*; *Figure 3—figure supplement 1B*), demonstrating that highly variable promoters are more cell-type and tissue specific in their expression.

The restricted expression (*Figure 3—figure supplement 1B*), biological processes (*Figure 3A*), and promoter TF grammar (*Figure 2D and E*) of genes associated with highly variable promoters led us to hypothesize that these are more prone to respond to external stimuli. Tumor necrosis factor (TNFα) induces an acute and time-limited gene response to NFkB signaling (*Nelson et al., 2004*; *Turner et al., 2010*), with negligible impact on chromatin topology (*Jin et al., 2013*), and is therefore suitable to study gene responsiveness. We profiled GM12878 TSSs and promoter activities with CAGE before and after 6 hr treatment with TNFα (*Supplementary file 4*). This revealed enrichment of up-regulated promoters among highly variable promoters (odds ratio (OR)=1.529, $p=4.563 \times 10^{-8}$, Fisher's exact test) as posited, while low variable promoters were mostly unaffected or down-regulated (OR = 0.459, $p<2.2 \times 10^{-16}$, Fisher's exact test; *Figure 3C*). In addition, low variable promoters had a weaker response (*Figure 3D*).

Furthermore, we observed drug-target genes and genes with GWAS hits to be regulated by highly variable promoters but essential genes to be regulated by low variable promoters (*Figure 3E*). In contrast, when we compared promoter expression between these same groups of genes we observed no association with drug-targets or GWAS-associated genes. Although essential genes are associated with higher promoter expression, this association is comparably weaker than that with promoter variability (*Figure 3—figure supplement 1C*).

Taken together, our results demonstrate the importance of low promoter variability for cell viability and survival in different conditions and reveal the responsiveness of highly variable promoters. They further indicate that the variability observed in promoter activity across individuals is strongly associated with the regulation of its associated gene, the expression breadth across cell types, and to some extent also the transcriptional noise across single cells, which reflects a selective trade-off between high stability and high responsiveness and specificity.

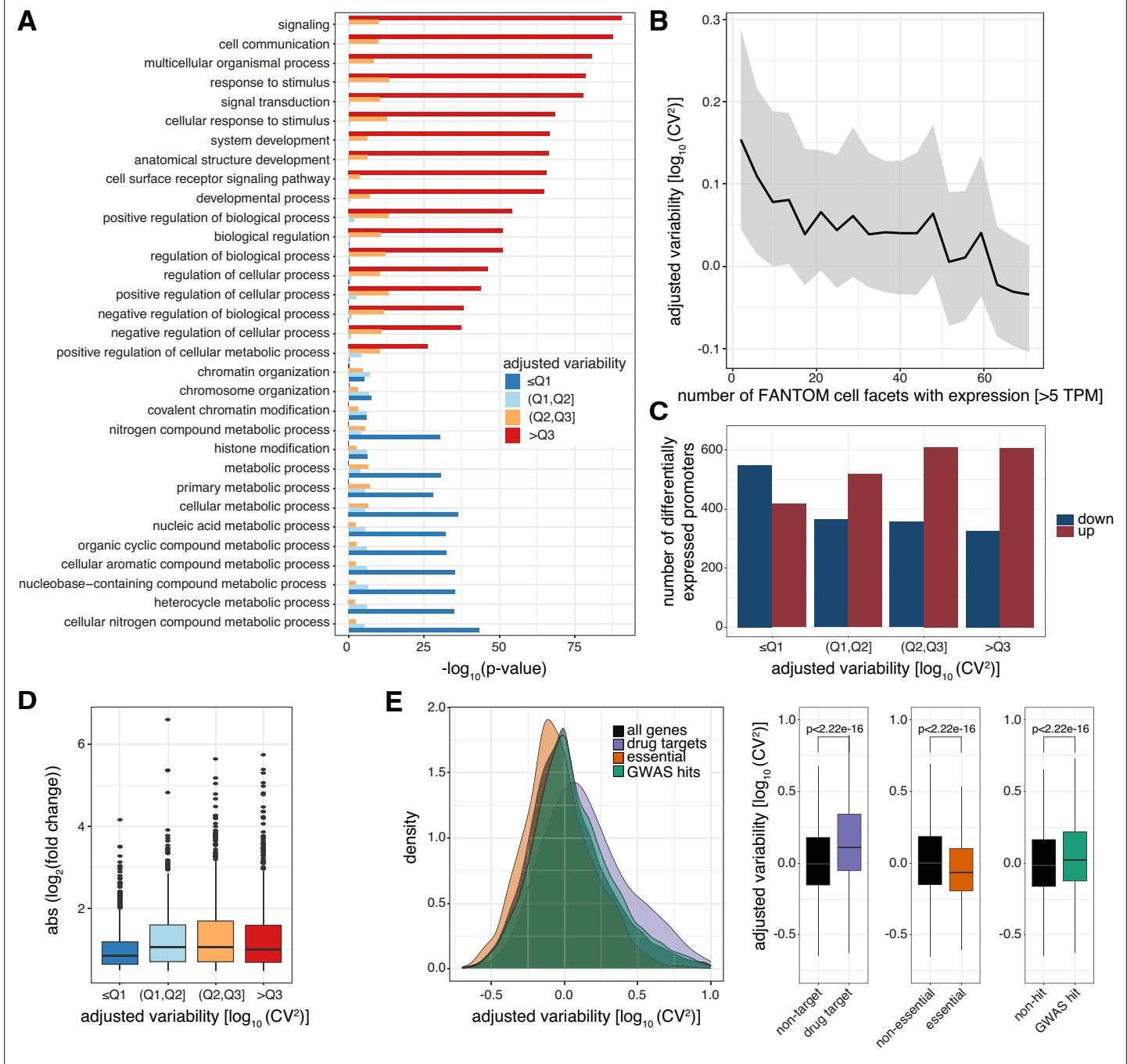

**Figure 3.** Levels of promoter variability are reflective of distinct biological processes and a selective trade-off between robustness and plasticity. (**A**) GO term enrichment, for biological processes, of genes split by associated promoter variability quartiles (**Q1, Q2, Q3**). Top 10 GO terms of all groups are displayed and ranked based on p-values of the >Q3 variability group. (**B**) Median promoter variability (line) and interquartile range (shading), as a function of the number of FANTOM cell facets (grouping of FANTOM CAGE libraries associated with the same Cell Ontology term) that the associated gene is expressed in (mean facet expression >5 TPM). (**C**) The number of differentially expressed promoters, split by variability quartiles, after 6 h TNFα treatment. Promoters are separated into down-regulated (blue) and up-regulated (red). p-values were calculated using Fisher's exact test. (**D**) Absolute log₂ fold change of differentially expressed promoters, split by variability quartiles, after 6 h of TNFα treatment. (**E**) Distribution of promoter variability associated with drug-targets (purple), essential (orange), or GWAS hits (green) genes, compared to all promoters (black). Left: density plot of promoter variability per gene category. Right: Box-and-whisker plots of promoter variability split by each category of genes. p-values were determined using the Wilcoxon rank-sum test. For all box-and-whisker plots, central band: median; boundaries: first and third quartiles; whiskers:+/-1.5 IQR.

The online version of this article includes the following figure supplement(s) for figure 3:

**Figure supplement 1.** Levels of promoter variability are reflective of distinct biological processes.

## Promoters with low variability have flexible transcription initiation architectures

Promoters are associated with different levels of spread of their TSS locations, which has led to their classification into broad or narrow (sharp) promoters according to their positional width (*Akalin et al., 2009*; *Carninci et al., 2006*; *Lehner, 2008*). Although the shape and distinct biological mechanisms of these promoter classes, for example, housekeeping activities of broad promoters, are conserved across species (*Carninci et al., 2006*; *Hoskins et al., 2011*), the necessity for positional dispersion of TSSs and its association with promoter variability are poorly understood.

Analysis of promoter widths revealed only a weak relationship with promoter variability. We observed an enrichment of highly variable promoters within narrow promoters having an interquartile range (IQR) of their CAGE-inferred TSSs within a width of 1–5 bp (p<2.2 × 10⁻¹⁶, OR = 2.04, Fisher's exact test). Low variable promoters, on the other hand, were enriched among those of size 10–25 bp (p<2.2 × 10⁻¹⁶, OR = 1.44, Fisher's exact test), but beyond this width the association is lost (*Figure 4—figure supplement 1A*). To simultaneously capture the spread of TSSs and their relative frequencies compared to total RNA expression within a promoter, we considered a width-normalized Shannon entropy as a measure of TSS positional dispersion (*Hoskins et al., 2011*). This measure will discern promoters whose relative TSS expression is concentrated to a small subset of their widths (low entropy) from those with a more even spread (high entropy). We observed that low variable promoters are associated with a higher entropy than promoters with high variability (*Figure 4A*). Consistently, low variable promoters tend to have more TSSs substantially contributing to their overall expression across individuals (*Figure 4—figure supplement 1B*). We reasoned that a weaker association between low promoter variability and broad width than with high entropy may be due to low variable promoters being composed of multiple clusters of TSSs (multi-modal peaks) from independent core promoters. Indeed, decomposition of multi-modal peaks within the CAGE TSS signals of promoters (*Supplementary file 5*) demonstrated that higher entropy reflects an increased number of decomposed promoters, as indicated by their number of local maxima of CAGE signals (*Figure 4B*).

The decomposed promoters of gene *UFSP2* (*Figure 4C and E*) clearly illustrate that the activity of sub-clusters of TSSs within promoters and their contributions to the overall activity of the encompassing promoter can vary to a great extent between individuals. In contrast, the decomposed promoters of gene *RIT1* (*Figure 4D and F*) contribute equally to the overall activity of the encompassing promoter across individuals. To assess in general how individual decomposed promoters influence the overall promoter variability, we calculated the expression-adjusted dispersion (adjusted log₁₀-transformed CV²) of local-maxima decomposed promoters. Interestingly, many of the decomposed promoters showed a vastly different variability across individuals compared to the promoters they originate from (*Figure 4—figure supplement 1C*). This disagreement indicates that decomposed promoters within the same promoter reflect core promoters that may either operate and be regulated independently of each other or differ in their ability to compete for the transcriptional machinery, both of which may contribute to the overall robustness or plasticity of the promoter and, in turn, the gene. As highly multimodal peaks are mainly found to be associated with low variable promoters, we hypothesized that this flexibility in core promoter usage may act as a compensatory mechanism to stabilize their expression.

If the effects of significant changes in expression across the panel are masked by compensatory changes in decomposed promoter usage within the same promoter, this would be revealed by low or even negative expression correlation between decomposed promoters (e.g., decomposed promoters 1 and 2 of *UFSP2*, *Figure 4C and E*). Indeed, we observed a strong association between promoter variability and the minimal expression correlation between decomposed promoter pairs within a promoter (*Figure 4G*). Low variable multi-modal promoters are associated with weakly or even negatively correlated pairs of decomposed promoters. In contrast, highly variable multi-modal promoters are associated with moderately or highly correlated pairs of decomposed promoters. The weak expression correlation between decomposed promoters of low variable promoters demonstrates that decomposed promoters may operate independently of each other, while negatively correlated pairs indicate a competition for the transcriptional machinery or a compensatory shift between decomposed promoters. The association between decomposed promoter correlation and overall promoter stability was maintained when all decomposed promoter pairs were considered (*Figure 4—figure supplement 1D*), and could not be explained by CpG island status (*Figure 4—figure supplement*

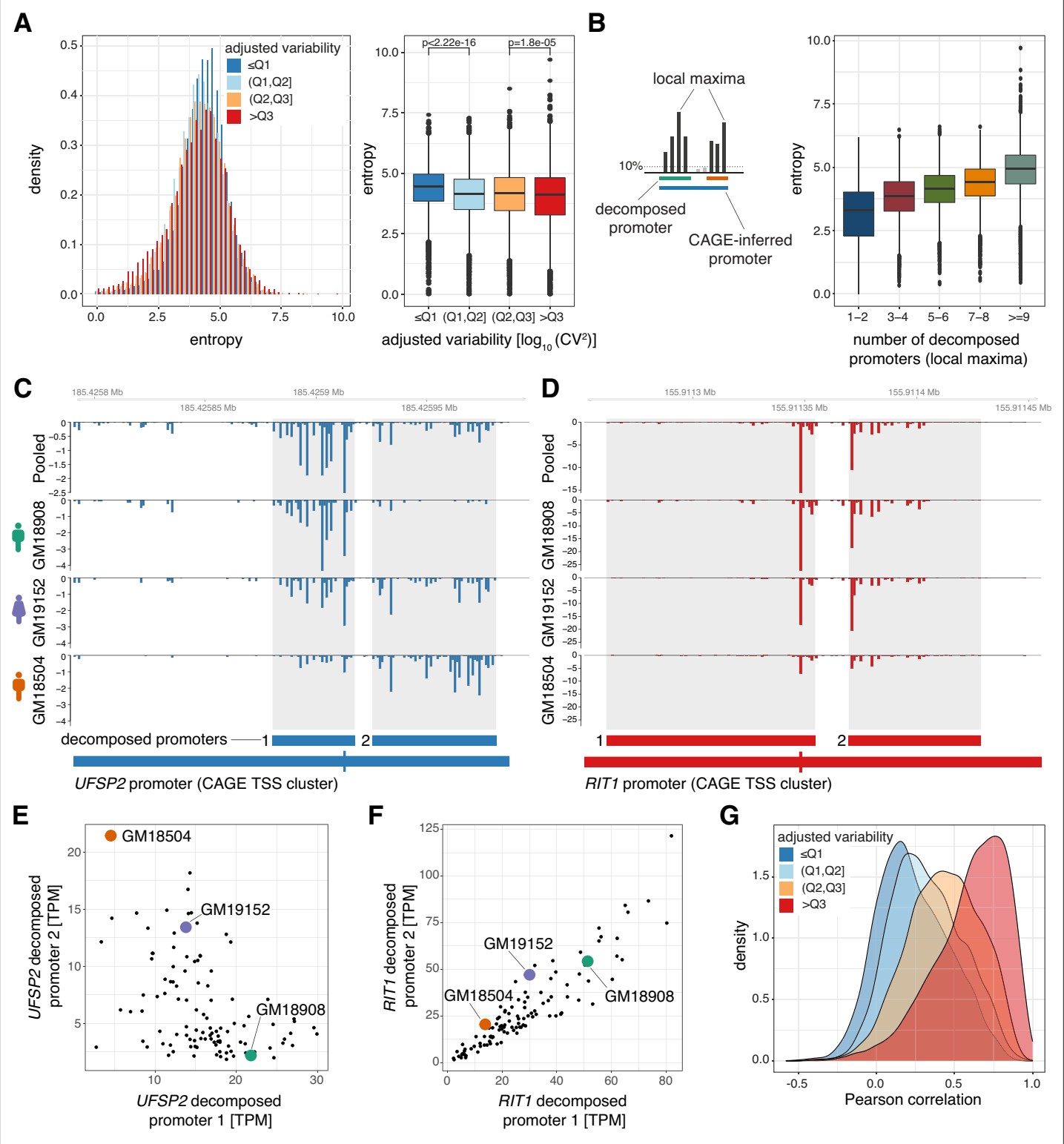

**Figure 4.** Low variable promoters exhibit flexibility in transcription initiation architecture. (**A**) Promoter shape entropy for promoters split by variability quartiles, displayed as densities (left) and in a box-and-whisker plot (right). (**B**) Illustration of the local maxima decomposition approach (left; see Materials and methods) and box-and-whisker plot displaying the relationship between the Shannon entropy and the number of local maxima-inferred decomposed promoters. (**C–D**) Examples of two promoters each containing two decomposed promoters exhibiting low correlation across individuals (panel C, gene UFSP2) and high correlation across individuals (panel D, gene RIT1). Both panels display genome tracks of average, TPM-normalized CAGE-inferred TSS expression levels across the panel (Pooled, top track) and for three individuals (GM18908, GM19152, GM18504, lower tracks). Below

*Figure 4 continued on next page*

*Figure 4 continued*

the genome tracks, the original promoter and resulting decomposed promoters (shaded in genome tracks) are shown. (**E–F**) Relationship between TPM-normalized CAGE expression of decomposed promoter 1 (x-axis) and 2 (y-axis) across all 108 LCLs for example genes *UFSP2* (**E**) and *RIT1* (**F**). The expression values for individuals included in panels B and C are highlighted. (**G**) Densities of the lowest Pearson correlation between all pairs of decomposed promoters originating from the same promoter across all CAGE-inferred promoters with at least two decomposed promoters. For all box-and-whisker plots, central band: median; boundaries: first and third quartiles; whiskers:+/-1.5 IQR.

The online version of this article includes the following figure supplement(s) for figure 4:

**Figure supplement 1.** Low variable promoters tend to be composed of multiple clusters of TSSs rather than broader TSS signatures.

**Figure supplement 2.** Low variable promoters with highly correlated decomposed promoters tend to have a fixed +1 nucleosome position across individuals.

*1E*). However, decomposed promoter expression correlation was associated with promoter width (*Figure 4—figure supplement 1F*), demonstrating that complex low variable promoters with multiple decomposed promoters require larger promoter width, while broad promoter width does not necessarily lead to lower promoter variability.

The spread and dominant position of TSSs in broad promoters are tightly linked to immediate downstream (+1) nucleosome positioning, and changes in +1 nucleosome positioning can alter the preferred TSS (*Dreos et al., 2016*; *FANTOM Consortium and the RIKEN PMI and CLST (DGT) et al., 2014*; *Haberle et al., 2014*). Hence, the variability and multi-modal TSS patterns of promoters could be related to their nucleosomal architectures. Indeed, comparison of +1 nucleosome positioning relative to the dominant TSS position of promoters across the panel revealed that low variable promoters tend to have stronger +1 nucleosome positioning (*Figure 4—figure supplement 2A and B*). However, when analyzing specifically multi-modal low variable promoters (containing at least two decomposed promoters), a strong +1 nucleosome was only observed for promoters with highly correlated decomposed promoters (*Figure 4—figure supplement 2C and D*). Furthermore, highly variable multi-modal promoters and those containing low correlated pairs of decomposed promoters exhibited more fuzzy +1 nucleosome positioning across cells (*Figure 4—figure supplement 2E and F*). Our results thus demonstrate that low variable promoters with flexible TSS usage, that is, having weakly or negatively correlated decomposed promoters, are characterized by less restrictive and more fuzzy +1 nucleosome positioning.

Taken together, our results demonstrate that flexible usage of core promoters within promoters with permissive nucleosomal architectures provide stability to the overall expression of a large subset of gene promoters with low variability.

## Alternative TSSs of low variability promoters indicate a novel mechanism of mutational robustness

While genetic variants associated with gene expression levels (expression quantitative trait loci, eQTLs) frequently occur within gene promoters, they are rarely found associated with housekeeping or ubiquitously expressed genes, and when they are, they have limited effect sizes (*Battle et al., 2017*). One explanation for this observation is that mutations that would significantly alter the expression of such genes would be detrimental to cell viability. In addition, the rare and limited effects of eQTLs on housekeeping genes might be due to mechanisms promoting mutational robustness. Our results (*Figure 4*) indicate that a flexible TSS architecture within a promoter may provide such a mechanism and thereby mask the effects genetic variants may have on individual decomposed promoters.

To test if flexibility in core promoter usage within a promoter may cause mutational robustness, we first performed local eQTL analysis on promoters (within 25 kb). We tested both the association between the genotypes of common genetic variants (MAF ≥10%) and the expression of promoters (promoter eQTL, prQTLs; *Figure 5A*, top) as well as those of decomposed promoters (decomposed promoter eQTL, dprQTL). 2,457 promoters were associated with at least one prQTLs (5% FDR; *Supplementary file 6*). While prQTLs were observed across all levels of promoter variability, they were more commonly associated with highly variable promoters (*Figure 5B*). Fewer prQTL single nucleotide polymorphisms (SNPs) and, in general, common variants were found proximal to low variable promoters, indicating a negative selection for these. As expected, the effect size for the most significant prQTL variant (lead SNP) for each promoter was positively associated with the expression variability of the

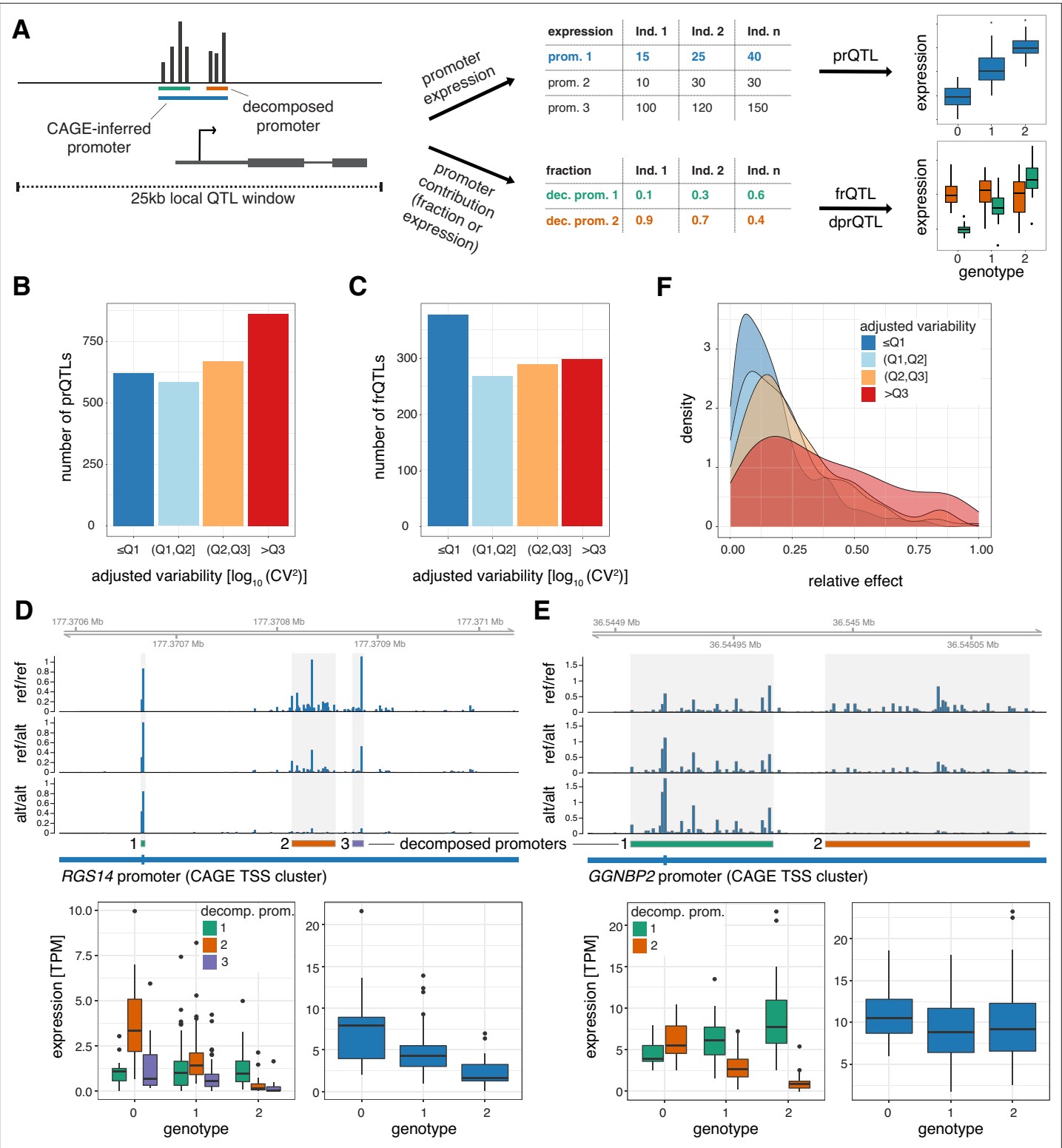

**Figure 5.** Plasticity in TSS usage is linked with increased mutational robustness. (**A**) Illustration of the strategy for testing the effects of genetic variants on promoter expression (prQTLs, TPM-normalized CAGE counts), decomposed promoter expression (dprQTLs, TPM-normalized CAGE counts), and decomposed promoter contribution to the encompassing promoter expression (frQTLs, ratios of TPM-normalized CAGE counts between decomposed and encompassing promoters). For both approaches only SNPs within 25 kb of the promoter CAGE signal summit were tested. (**B**) Number of prQTLs detected (FDR < 0.05), split by promoter variability quartiles. (**C**) Number of encompassing promoters with at least one frQTL detected for a contained decomposed promoter (FDR < 0.05), split by encompassing promoter variability quartiles. (**D–E**) Examples of two promoters associated with frQTLs for

*Figure 5 continued on next page*

*Figure 5 continued*

a highly variable promoter with limited buffering of promoter expression (panel D, gene *RGS14*) and for a low variable promoter with strong buffering of promoter expression (panel E, gene *GGNBP2*). Upper panels display genome tracks showing average TPM-normalized CAGE data across homozygous individuals for the reference allele (top track), heterozygous individuals (middle track), and homozygous individuals for the variant (alternative) allele (bottom track). The bottom left subpanels display box-and-whisker plots of the differences in TPM-normalized CAGE data between genotypes for each decomposed promoter. The bottom right subpanels display box-and-whisker plots of the differences in TPM-normalized CAGE data between the three genotypes for the original encompassing promoter. For all box-and-whisker plots, central band: median; boundaries: first and third quartiles; whiskers:+/-1.5 IQR. (F) Density plot of the maximal relative change in expression between reference and variant alleles (relative effect size) for the most significant frQTL of each broad promoter with FDR ≥ 5%, split by variability quartiles.

The online version of this article includes the following figure supplement(s) for figure 5:

**Figure supplement 1.** Low variable promoters with highly correlated decomposed promoters tend to have a fixed +1 nucleosome position across individuals.

promoter (Spearman's rank correlation $\rho$ = 0.16, p< 2.2 × 10$^{-16}$, *Figure 5—figure supplement 1A*). This indicates that, in addition to having fewer proximal genetic variants, low variable promoters are less likely to have prQTLs with large regulatory effects. However, QTLs of decomposed promoter expression (dprQTLs) exhibited similar prevalence (*Figure 5—figure supplement 1C*) and maximum effect sizes (*Figure 5—figure supplement 1D*) across promoter variability classes. This suggests that a flexible TSS architecture can limit the impact of genetic variants on promoter expression.

To further investigate the observed disagreement between the variability of promoters and their decomposed promoters, we tested the association between the genotypes of common genetic variants and the contribution of decomposed promoters to the overall expression of their encompassing (non-decomposed) promoters (fraction eQTL, frQTL; *Figure 5A*, bottom). We identified 1,230 promoters to be associated with at least one frQTL (5% FDR; *Supplementary file 7*). Unlike the prQTLs and dprQTLs, the frQTLs were more commonly associated with decomposed promoters from low variable promoters (*Figure 5C*). Conceptually, the frQTLs can affect decomposed promoter usage and overall promoter expression levels to different degrees, as exemplified by the promoters of genes *RGS14* and *GGNBP2* (*Figure 5D and E*). Gene *RGS14* has three decomposed promoters localized within its promoter (*Figure 5D*), for which SNP rs56235845 (chr5:177371039 T/G) was strongly associated with the contribution to the overall promoter activity for only decomposed promoters 1 and 2 (frQTL beta = 0.210, –0.181, –0.062; FDR = 2.42 × 10$^{-5}$, 2.54×10$^{-8}$, 2.64×10$^{-2}$, for decomposed promoters 1, 2, and 3, respectively). Despite the limited association of the variant with decomposed promoter 3, it still had a noticeable association with the overall promoter activity (prQTL beta = −2.47, FDR = 3.57 × 10$^{-5}$; *Figure 5D*, bottom right). In contrast, SNP rs9906189 (chr17:36549567 G/A) was strongly associated with the contribution to the overall promoter activity for both decomposed promoters of gene *GGNBP2* (frQTL beta = 0.222,–0.222; FDR = 2.05 × 10$^{-26}$, 2.05×10$^{-25}$, for decomposed promoters 1 and 2, respectively), but in opposite directions (*Figure 5E*). Interestingly, this switch in decomposed promoter usage translates into a limited effect on the overall promoter activity (prQTL beta = −0.063, FDR = 0.989; *Figure 5E*, bottom right).

Both examples, a partial shift (*Figure 5D*) and a switch (*Figure 5E*) in decomposed promoter usage, are indicative of plasticity in TSS usage, which can secure tolerable levels of steady-state mRNA. Although frQTLs were associated with promoters across the wide spectrum of promoter variabilities (*Figure 5C*), they showed a large difference in their relative effect on the overall promoter activity (maximal relative change in expression between reference and variant alleles; *Figure 5F*). frQTLs associated with highly variable promoters tend to have a larger relative effect on the overall promoter activity compared to frQTLs associated with low variable promoters. This association is further maintained at the gene level (adjusted RNA-seq *Lappalainen et al., 2013* CV$^2$; *Figure 5—figure supplement 1B*; *Supplementary file 8*), demonstrating that individual differences in decomposed promoter usage contribute to low promoter variability and, in turn, low gene variability. In total, we found 286 promoters (out of 1230) of 284 genes to be associated with stabilizing frQTLs, for which the same SNP was associated with at least two decomposed promoters (5% FDR) with relative effects in opposite directions (*Supplementary file 9*). Our results thus indicate that TSS usage flexibility confers mutational robustness that stabilizes the variability of promoters and their associated genes.

Taken together, integrating prQTLs, dprQTLs, and frQTLs provides novel insights into how common genetic variants can influence TSS usage in humans and its potential impact on gene expression. We

demonstrate that low variable promoters characterized by multiple decomposed promoters (multi-modal TSS usage) are less affected by the presence of genetic variants compared to highly variable promoters. In addition, we find that part of this tolerance can be explained by a, previously unreported, mechanism of mutational robustness through plasticity in TSS usage. The prevalence of expression-independent decomposed promoters within low variable promoters, as suggested by low pairwise correlation, indicates an extensive regulatory role of TSS plasticity in attenuating expression variability of essential genes.

## Discussion

In this study, we provide an extensive characterization of promoter-associated features influencing variability in promoter activity across human individuals and demonstrate their importance for determining stability, responsiveness, and specificity. Overall, we show that the local DNA sequence, putative TF binding sites, and transcription initiation architecture of promoters are highly predictive of promoter variability.

Although the classifier based on TF binding site sequence and CpG island status was able to predict promoter variability well (AUC = 0.79 on the test set), it did not perform as well as the CNN model (AUC = 0.84 on the test set), which was trained on DNA sequence alone. This indicates that additional information influencing variability may be encoded within the promoter sequence. For instance, the density and variations of Initiator elements within promoters could influence TSS flexibility (*Carninci et al., 2006*; *Frith et al., 2007*; *Haberle et al., 2014*; *Nepal et al., 2020*). In addition, di- or tri-nucleotide sequence patterns and stretches of high AT-richness, which influence local nucleosome positioning (*Dreos et al., 2016*; *Haberle et al., 2014*; *Segal et al., 2006*), impose different requirements for chromatin remodeling activities (*Lorch et al., 2014*) at gene promoters of low and high variability, which in turn may affect their variability and responsiveness. The promoter sequence may also encode a promoter's intrinsic enhancer responsiveness (*Arnold et al., 2017*), which may influence its expression variability. Although current data cannot distinguish between direct or secondary effects, an increased variability mediated via enhancers is supported by a higher dependency on enhancer-promoter interactions for cell-type-specific genes compared to housekeeping genes (*Furlong and Levine, 2018*; *Schoenfelder and Fraser, 2019*). However, compatibility differences between human promoter classes and enhancers only result in subtle effects in vitro (*Bergman et al., 2022*), suggesting that measurable promoter variability is likely a result of both intrinsic promoter variability and additive or synergistic contributions from enhancers. Directly modeling the influence and context-dependency of enhancers on promoter variability would therefore be important to further characterize regulatory features that may amplify gene expression variability.

Despite a clear association with high promoter CpG content and housekeeping genes, low variable promoters were not strongly associated with a broader width, which we would expect from promoters in CpG islands and with housekeeping activity (*Carninci et al., 2006*). Rather, our results suggest that low variability requires a certain minimum promoter width that can encompass a transcription initiation architecture competent of attenuating variability through flexible TSS usage. Switching between proximal clusters of TSSs (decomposed promoters) within a larger promoter is fundamentally different from that between alternative promoters (*Garieri et al., 2017*; *Valen et al., 2008*; *Zhang et al., 2017*), which will more likely lead to differences in transcript and protein isoforms. Rather, a flexible initiation architecture enables several points of entries for RNA polymerase II to initiate in the same promoter. This ensures proper gene expression across different cell types (*FANTOM Consortium and the RIKEN PMI and CLST (DGT) et al., 2014*; *Kawaji et al., 2006*) and developmental stages (*Haberle et al., 2014*). Interestingly, ETS factors, here associated with low variable promoters, can interact with transcriptional co-activators and chromatin modifying complexes (*Curina et al., 2017*; *Göös et al., 2022*). ETS factors may therefore play a role in TSS selection in promoters with multi-modal architectures (*Lam et al., 2019*). Here, we show that such flexibility is also associated with low variability across individuals for the same cell type. Our findings further indicate that plasticity in TSS usage within a promoter confers a, previously unreported, layer of mutational robustness that can buffer the effects of genetic variants, leading to limited or no impact on the overall promoter expression. Of note, the presence of weak or negatively correlated expression patterns between decomposed promoters for a large number of promoters suggests that such buffering events will be revealed for more genes with an increased sample size.

Flexibility in TSS usage may ensure transcriptional robustness of genes both in different environments and in the face of genetic variation. Since promoter shape is generally conserved across orthologous promoters (*Carninci et al., 2006*; *Hoskins et al., 2011*), it is plausible that robustness through flexible TSS usage is a conserved mechanism. In support, genetic variants may affect promoter shape for ubiquitously expressed genes in flies with limited effect on promoter expression (*Schor et al., 2017*). Changes in promoter shape in flies thus likely recapitulates the plasticity in TSS usage across human LCLs, despite apparent differences in core promoter elements, promoter nucleotide content, and regulatory features between flies and humans.

Notably, many of the promoter features we, and others (*Sigalova et al., 2020*), have identified to be predictive of promoter variability, including the presence or absence of CpG islands and TATA boxes, have previously been linked with different levels of transcriptional noise as inferred from single-cell experiments (*Faure et al., 2017*; *Morgan and Marioni, 2018*). This suggests that variability in promoter activity across individuals partly reflects the stochasticity in gene expression across cells. Given that the underlying sources of variation are different, for example, genetic and environmental versus stochastic, this indicates that mechanisms that contribute to the buffering of stochastic noise at a single cell level can also contribute to the attenuation of genetic and environmental variation at an individual level.

We observed less restrictive +1 nucleosome positioning across individuals at low variable promoters with flexible TSS usage. Notably, these promoters are also associated with a fuzzy +1 nucleosome positioning across cells within an individual. This indicates that positional shifts of nucleosomes in coordination with shifts in core promoter usage may be due to an inherent property of these promoters in addition to the influence by genetic variation. Our observations indicate that multiple configurations of accessible chromatin may exist for low variable promoters across single cells, which may cause stochastic TSS selection with no or only low impact on expression level. This is compatible with a high density of pyrimidine/purine (YR) dinucleotides within broad promoters (*Carninci et al., 2006*; *Frith et al., 2007*), which provide a flexibility for transcription initiation sites in the absence of strong positional signals like the TATA box (*Carninci et al., 2006*; *FANTOM Consortium and the RIKEN PMI and CLST (DGT) et al., 2014*; *Müller and Tora, 2014*). Genetic variants biasing such TSS selection and the preference for any open chromatin configuration may therefore cause observable shifts in TSS usage between individuals, enabled by the flexible nucleosome and TSS architecture of the promoter.

It is important to note that the regulatory programs of EBV-immortalized LCLs, like other cultured cells, have been shown to be susceptible to genotype-independent sources of variation, such as

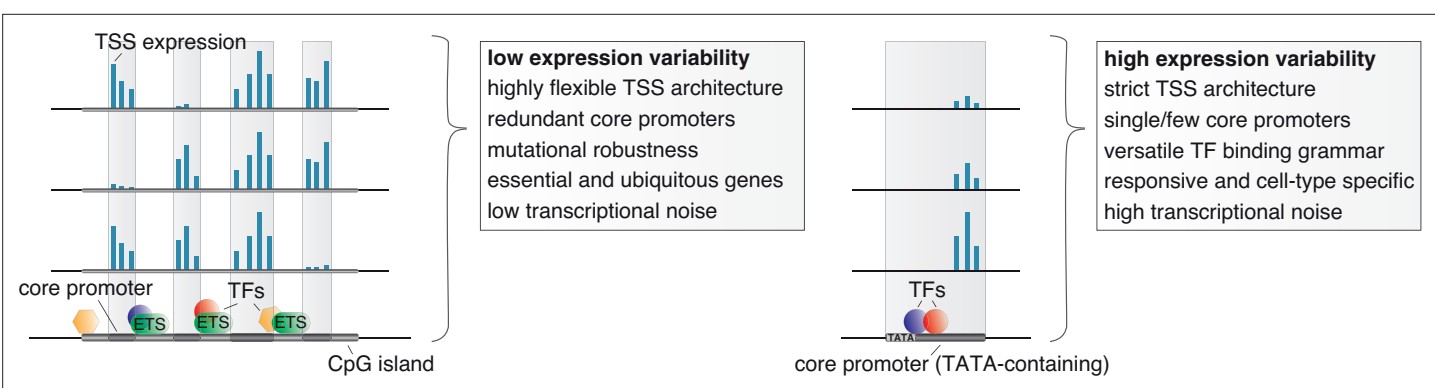

**Figure 6.** Unifying mechanisms influencing the variability in expression across individuals, the specificity in expression across cell types, and the stochasticity in expression across individual cells. Low variable promoters (left) are frequently associated with high CpG content (CpG islands), multiple binding sites of ETS factors, and a highly flexible transcription initiation architecture arising from multiple redundant core promoters (decomposed promoters) in a permissive nucleosome positioning environment. These stabilizing features along with a less complex TF binding grammar likely also act to buffer transcriptional noise across single cells and cause ubiquitous expression across cell types. The flexibility in redundant core promoter activities confers a novel layer of mutational robustness to genes. Highly variable promoters (right), on the other hand, are associated with a highly versatile TF regulatory grammar, TATA boxes, and low flexibility in TSS usage. These features likely cause, in addition to high expression variability between individuals, a responsiveness to external stimuli, cell-type restricted activity, high transcriptional noise across single cells, and less tolerance for genetic variants.

primary cellular heterogeneity and EBV's viral cellular reprogramming (*Choy et al., 2008*; *Ozgyin et al., 2019*). While we cannot exclude that such sources may influence the measured variability of some genes, we speculate that non-genetic variation would rather dampen than elevate the associations and effect sizes we report here, in particular the decomposed promoter switches highlighted by our genotype association analysis.

Taken together, our results favor a model in which the regulation of transcriptional noise across single cells reflects specificity across cell types and dispersion across individuals with shared mechanisms conferring stochastic, genetic and environmental robustness (*Figure 6*). There are several implications of this model. First, the link between low transcriptional noise and low individual variability of promoters and their associations with ubiquitous and essential genes indicates that regulatory mechanisms that ensure broad expression across cell types may enforce low variability across individuals and single cells. Second, our results indicate that encoding responsiveness or developmentally restricted expression patterns of gene promoters may require high stochasticity in expression across single cells, which in turn may disallow ubiquitous expression across cell types. Thus, it is likely that increased variability is not just reflecting the absence of regulatory mechanisms that attenuate variability but the presence of those that amplify it. Finally, given that mutational robustness through flexible TSS usage is mostly associated with low variable genes, this implies that cell-type restricted, responsive and developmental genes may be more susceptible to trait-associated genetic variants, which finds support in the literature (*Finucane et al., 2015*; *Roadmap Epigenomics Consortium et al., 2015*; *Timshel et al., 2020*).

## Materials and methods
### LCL cell culturing
Epstein-Barr virus immortalized LCLs (*Supplementary file 1*) were obtained from the NIGMS Human Genetic Cell Repository at Coriell Institute for Medical Research. Cells were incubated at 37°C at 5% carbon dioxide in the Roswell Park Memorial Institute (RPMI) Medium 1640 supplemented with 2mM L-glutamine and 20% of non-inactivated fetal bovine serum and antibiotics. Cell cultures were split every few days for maintenance. All 108 LCLs were grown unsynchronized for 5–7 passages and harvested when they reached>20million cells. The Coriell Institute for Medical Research frequently screens cells for mycoplasma contamination and verifies cell line identity. All cell culturing was done within a year of purchase. To confirm that samples were free of mycoplasma contamination, CAGE reads were aligned to 4 mycoplasma reference genomes, including *Mycoplasma hominis*, which showed no indication of contamination for any sample (*Olarerin-George and Hogenesch, 2015*).

### CAGE library preparation, sequencing, and mapping
CAGE libraries were prepared in 10 batches in total as described elsewhere (*Andersson et al., 2014b*; *Takahashi et al., 2012*) from 1500 ng total RNA from each LCL. 23 libraries (*Supplementary file 1*) underwent a second round of size selection (Invitrogen E-Gel) to remove excessive primer dimers. The libraries were quality checked using an Agilent 2100 Bioanalyzer system with a RNA pico chips kit (Agilent) and quantified using DNA 1000 chips kit (Agilent). Pooled libraries (*Supplementary file 1*) were sequenced with spiked-in PhiX on an Illumina HiSeq 2500 machine single-end for 50 cycles using v4 sequencing chemistry (Illumina Inc) and a custom sequencing primer (*Takahashi et al., 2012*). Libraries were split by barcode and reads were trimmed to remove linker sequences and filtered for a minimum sequence quality of Q30 in 50% of base pairs using the FASTX-Toolkit. rRNA reads matching subsequences of the human ribosomal DNA complete repeating unit (U13369.1) were removed using rRNAdust (version 1.06) (*FANTOM Consortium and the RIKEN PMI and CLST (DGT) et al., 2014*). Mapping to the human reference genome (hg38) was performed using BWA (version 0.7.15-r1140) allowing a maximum edit distance of 2. To reduce mapping bias, reads were re-mapped using the WASP pipeline (*van de Geijn et al., 2015*) and BWA, taking into account biallelic SNVs (*Lowy-Gallego et al., 2019*). Reads with a mapping quality of 20 were retained for further analyses. Sample-related information, including CAGE run batch ID and E-gel information, are provided in *Supplementary file 1*.

## CAGE tag clustering, filtering and quantification

CAGE-defined transcription start sites (CTSSs) were identified from 5′ ends of mapped CAGE reads for each strand separately. The expression of CTSSs for each LCL was quantified from the number of CAGE reads sharing 5′ ends using CAGEfightR (version 1.10) (*Thodberg et al., 2019*). To identify broad promoters that could potentially encompass multiple alternative core promoters (decomposed promoters), we performed lenient positional clustering of CTSSs (tag clustering) (*Carninci et al., 2006*), grouping CTSSs on the same strand within 60 bp of each other. To exclude rare promoters within the panel, tag clustering was performed on CTSSs with at least 1 CAGE read in at least 5 LCLs. The expression of each tag cluster (CAGE-inferred promoter) in each individual LCL was quantified by aggregating the expression of all CTSSs falling within the defined tag cluster region. To allow capture of flexible TSS usage within promoters across the panel, no support filtering was performed at CTSS level for expression quantification. Expression levels were converted to tags per million (TPM), by normalizing the expression count of each tag cluster in each library as a fraction of its number of mapped CAGE reads, scaled by $10^6$. Tag clusters were filtered to be proximal to GENCODE-annotated TSSs (hg38, version 29, within 1000 bp upstream) and to have at least 10 read counts in more than 10 LCLs. The resulting 29,001 gene-associated CAGE-inferred promoters were later decomposed by local maxima decomposition to split multi-modal tag clusters (https://github.com/anderssonlab/CAGEfightR_extensions, version 0.1.1; *Andersson, 2021*). First, for each CAGE-inferred promoter, local maxima of within-promoter CTSSs with the highest pooled expression separated by at least 20 bp were identified. Second, decomposition was performed for each local maxima separately in decreasing order of pooled expression level. For each local maxima, the fraction between the pooled expression of each CTSS to that of the local maxima was calculated. All CTSSs associated with at least 10% of the local maxima signal that were not gapped by more than 10 bp with CTSS expression less than this value were retained in a new decomposed promoter. For smoothing purposes, neighboring non-zero CTSSs within 1 bp distance of CTSSs fulfilling the fraction criterion were also included. Subsequently, decomposed promoters were merged if positioned within 1 bp from each other.

## Geuvadis YRI RNA-seq data analysis

Gene expression data quantified in the recount2 project (*Collado-Torres et al., 2017*) using Geuvadis YRI RNA-seq data (*Lappalainen et al., 2013*) was downloaded using the recount R package. Only genes with more than 1 transcript per million in at least 10% of YRI samples were considered for expression variability calculation.

## GM12878 scRNA-seq data analysis

GM12878 10 X Genomics scRNA-seq data (*Osorio et al., 2019*) was downloaded from Gene Expression Omnibus (GSE126321) and processed using Seurat (version 4.0.3) (*Hao et al., 2021*). Cells with a proportion of mitochondrial reads lower than 10% and a sequencing depth deviating less than 2.5 times the standard deviation from the average sequencing depth across cells were considered. The expression of genes with read counts observed in at least 10 cells were normalized using scran (version 1.18.7) (*Lun et al., 2016*) and retained for expression variability calculation.

## Measuring expression variability across individuals

The raw dispersion of each CAGE tag cluster was calculated using the squared coefficient of variation ($CV^2$) of TPM-normalized promoter (or decomposed promoter) expression across the LCL panel and subsequently $\log_{10}$-transformed. Adjustment of the mean expression-dispersion relationship was performed by subtracting the expected $\log_{10}$-transformed dispersion for each promoter according to its expression level, using a running median (width 50, step size 25) of raw dispersions ($\log_{10} CV^2$) ordered by mean expression level (TPM) across the panel, as described elsewhere (*Kolodziejczyk et al., 2015*; *Newman et al., 2006*). The same strategy was used to calculate the adjusted dispersion of gene expression from RNA-seq and scRNA-seq data. Promoters were grouped by variability according to the quartiles of expression-adjusted dispersions (≤Q1, (Q1, Q2], (Q2, Q3],>Q3).

## Neural network model, training and hyperparameter tuning

A simple neural network architecture was designed to learn to predict low and high variability from DNA sequence. The neural network model uses as input one-hot-encoded DNA sequences (A =

[1,0,0,0], C = [0,1,0,0], G = [0,0,1,0], T = [0,0,0,1]) from the human reference genome (hg38) to predict low and highly variable promoter activity as output. Although CAGE-inferred promoters varied in width, we made use of fixed-length 600bp sequences for each promoter centered on its pooled CAGE summit CTSS. 600bp was used to make sure that sequences influencing promoter variability contained within regions that could cover a central open chromatin site (150–300bp) as well as within flanking nucleosomal DNA (150–200bp) were captured, where also most of the expression output of a promoter originate (*FANTOM Consortium and the RIKEN PMI and CLST (DGT) et al., 2014*).

The model (*Figure 2—figure supplement 1A*) consists of one convolutional layer with 128 hidden units and a kernel size of 10, followed by global average pooling and two dense layers with 128 and 2 nodes, respectively. Batch normalization and dropout (0.1) were applied after each layer. The ReLU activation function (*Agarap, 2019*) was used in all layers except the final layer, in which a sigmoid activation function was used to predict the variability class (low or high adjusted dispersion).

Promoter sequences from chromosomes 2 and 3 were used as the test set and those from the remaining chromosomes were used for training and hyperparameter tuning with a fivefold cross-validation. Hyperparameters were manually adjusted to yield the best performance on the validation set. The neural network model was implemented and trained in Keras (version 2.3.0, https://github.com/keras-team/kera *Chollet, 2022*) with TensorFlow backend (version 1.14) (*Abadi et al., 2016*) using the Adam optimizer (*Kingma and Ba, 2017*) with a learning rate of 0.0001, batch size of 64, and early stopping with the patience of 15 epochs.

We initially used the first and third quartiles (Q1 and Q3) to distinguish low variable promoters (≤Q1) from highly variable promoters (>Q3), corresponding to an adjusted $\log_{10}$-transformed $CV^2$ of –0.1490 and 0.1922, respectively. To reduce false positives, we slightly adjusted the thresholds for low and highly variable promoters to –0.20 and 0.25, respectively. The final training and test sets for the neural network model together consisted of 5054 low variable and 5683 highly variable promoters. To ensure consistency, the same thresholds were used for training and testing with Random Forest and decision tree classifiers (see below).

## Motif discovery using DeepLIFT and TF–MoDISco

To interpret the neural network model we used DeepLIFT (*Shrikumar et al., 2019*), a feature attribution method, to compute importance scores for each nucleotide in the 600 bp input sequences for low and highly variable promoters. DeepLIFT relies on backpropagation of the contributions of all neurons in the neural network to the input features, nucleotides, and was used to estimate the importance of each position and nucleotide in the input sequences to predict high and low variability. The resulting importance scores were supplied to TF-MoDISco (Transcription Factor Motif Discovery from Importance Scores) (*Shrikumar, 2020*) to identify DNA stretches (seqlets) with high importance for the predictions. DeepLIFT and TF-MoDISco were run independently on the input sequences for low variable and highly variable promoters. TF-MoDISco identified 18,035 seqlets for low variable promoters and 21,942 seqlets for highly variable promoters by using the importance scores from DeepLIFT over a width of 15 bp with a flank size of 5 bp and a FDR threshold of 0.05. The seqlets identified were merged in 41 and 47 metaclusters for low and highly variable promoters, respectively.

We used Tomtom (MEME package 5.1.1; *Gupta et al., 2007*) to match the resulting metaclusters to known TF motifs (in MEME format) from the JASPAR database (release 2020, hg38; *Fornes et al., 2020*). We compared each non-redundant JASPAR vertebrate frequency matrix with the metaclusters using Tomtom based on the Sandelin and Wasserman distance (*Sandelin and Wasserman, 2004*). Matches were considered those with a minimum overlap between query and target of 5 nucleotides and a p-value <0.05.

## Random forest, Boruta and SHAP analysis

To identify broad-scale trends of high CpG content, we calculated CpG observed/expected ratio in windows +/-500bp around the pooled summit CTSS of each promoter. Calculated CpG ratios revealed a bimodal distribution that informed on thresholding high CpG content promoters as those with CpG observed/expected ratio>0.5 (*Figure 2—figure supplement 2A*).

Predicted transcription factor binding sites for 746 TFs with scores of 500 or greater ($P<10^{-5}$) (hg38) were obtained from JASPAR (release 2020, hg38) (*Fornes et al., 2020*) and for each TF, presence/absence was obtained by overlapping predicted TF binding sites with promoters considered in the

modeling. Together, the CpG content status and the presence/absence of predicted TF binding sites were used as features for predicting high and low variability using Random Forest (*Pedregosa et al., 2011*).

Similarly to the neural network model, promoters from chromosomes 2 and 3 were only used as the test set. The remaining promoters were used for training and hyperparameter tuning with five-fold cross-validation. The Random Forest model was implemented and trained in Scikit-learn (version 2.3.0) with 500 trees, a maximum depth of trees of 10, 50samples split per node, and 50samples to be at a leaf node. The remaining hyperparameters were kept with default values.

Instead of selecting features directly from the Random Forest model, the BorutaShap package (*Keany, 2020*) was used for feature selection. The main advantage of using the Boruta approach is that the features compete with their randomized version (or shadow feature) and not with themselves. Therefore, a feature is considered relevant only if its score is higher than the best randomized feature. In this way, from the 746 original TF features, only 125 features were kept. The features were selected using only promoters from the training set. Finally, the SHAP library (*Lundberg and Lee, 2017*) was used to explain the importance of the 125 selected features for the two promoter classes. SHAP calculates Shapley values, a game theoretic approach for optimal credit allocation during cooperation, which can be used to estimate the marginal contribution of each feature to a model's predictions.

## Decision tree baseline model

To evaluate the contribution of TF binding site presence for predicting promoter variability, we trained a baseline model based on CpG content status and TATA-box presence only. CpG content status (CpG observed/expected ratio>0.5) and the presence/absence of predicted TBP binding sites were used as features for predicting high and low variability using a decision tree classifier. The decision tree model was implemented and trained in Scikit-learn (version 2.3.0) using default parameters. Training and test data were defined as for the CNN and Random Forest models.

## Tissue-, cell-type specificity and gene annotations

RNA-seq gene expression values across tissues were obtained from the GTEx consortium (*Battle et al., 2017*). Promoters were considered expressed in tissues in which their corresponding gene had≥5RPKM average expression across donors.

CAGE gene expression values across cell types were obtained from the FANTOM5 project (*FANTOM Consortium and the RIKEN PMI and CLST (DGT) et al., 2014*). The average normalized (tags per million, TPM) expression per gene was calculated across samples associated with the same cell type facet (grouping of CAGE libraries according to Cell Ontology annotation of samples), according to *Andersson et al., 2014a*, and a gene was considered expressed in a cell type facet if the average expression was≥5TPM.

Gene lists for FDA approved drug-targets (*Wishart et al., 2018*), essential genes (*Hart et al., 2017*) and GWAS hits (*MacArthur et al., 2017*) were downloaded from the MacArthur Lab Repository (https://github.com/macarthur-lab/gene_lists; *MacArthur, 2019*).

## GM12878 cell culturing, TNF-α stimulation and differential expression analysis

GM12878 cells were obtained from the NHGRI Sample Repository for Human Genetic Research at Coriell Institute for Medical Research. Unstimulated GM12878 cells and those stimulated with 25 ng/ml TNF-α for 6 hr were harvested with four replicates for each condition. Cell culturing, CAGE library preparation and mapping were done as described above for the LCL panel. CAGE reads supporting each of the final filtered promoters identified in the LCL panel were counted for each replicate using CAGEfightR (version 1.10; *Thodberg et al., 2019*). Differential expression analysis of the aggregated CTSS counts was performed using standard library size adjustment and a generalized linear model with DESeq2 (version 1.30.1; *Love et al., 2014*). Promoters with FDR-adjusted p-value ≤0.05 were considered to be differentially expressed.

## Correlation analysis of decomposed promoter expression

To test if decomposed promoters could act independently of each other, we calculated Pearson correlation coefficients of LCL expression between pairs of decomposed promoters originating from

the same promoter. We focused on promoters with at least 2 decomposed promoters significantly contributing to the overall expression of the promoter. Specifically, we considered decomposed promoters whose expression accounted for at least 5% of the overall promoter expression in at least half of all LCLs, resulting in 37,663 decomposed promoters of 14,889 promoters. To avoid potential bias introduced from a variable number of decomposed promoters per promoter, we considered the lowest correlation across decomposed promoter pairs within a promoter.

## Nucleosome positioning analysis

Micrococcal nuclease-digested nucleosome sequencing (MNase-seq) data from 7 EBV-immortalized LCLs (GM18507, GM18508, GM18516, GM18522, GM19193, GM19238, GM19239) were obtained from GEO (GSE36979) (*Gaffney et al., 2012*).

The locations and fuzziness scores of nucleosomes were called with DANPOS2 (version 2.2.2) (*Chen et al., 2013*) using the dpos command on each LCL separately. +1 nucleosomes were defined as the closest downstream nucleosome of the dominant TSS position (derived from pooled CAGE data) of each CAGE-inferred promoter.

Positional cross correlations were calculated between CAGE TSSs and 5' ends of MNase-seq reads on the same strand at dominant TSS positions of CAGE-inferred promoters ±500bp (maximum lag 250) to identify their most likely separation. Cross-correlation analysis was performed on either pooled CAGE data (across all 108 LCLs) versus pooled MNase-seq data (across all 7 LCLs) or using only CAGE and MNase data from one LCL (GM18516). Finally, for both analyses, a weight for each promoter was calculated from the geometric mean of aggregated MNase and CAGE signals. This was used to calculate a weighted average of the cross correlations across considered promoters.

## Mapping QTLs

We tested both the association between the genotypes of common genetic variants (MAF ≥10%) and the expression of promoters (promoter eQTL, prQTLs) and decomposed promoters (decomposed promoter eQTL, dprQTLs), as well as their association with the contribution of to the overall expression of the encompassing (non-decomposed) promoter (fraction eQTL, frQTL). prQTLs, dprQTLs, and frQTLs were mapped using the MatrixEQTL R package (version 2.3) (*Shabalin, 2012*). We controlled for genetic population stratification and library preparation batches (*Supplementary file 1*) by including these as covariates. In addition, we included the first 5 principal components derived from normalized promoter expression values (TPM) as covariates for prQTLs.

For prQTL detection, all 29,001 promoters were tested using TPM-normalized expression values. For frQTLs, we calculated the fractional contribution of each decomposed promoter to the expression of its original promoter. To focus the dprQTL and frQTL analyses on relevant shifts in TSS usage, we considered only decomposed promoters whose expression accounted for at least 5% of the overall promoter expression in at least half of all LCLs and promoters with at least 2 such decomposed promoters, resulting in 37,663 decomposed promoters of 14,889 promoters.

For each promoter, we tested common (minor allele frequency ≥10%) biallelic SNVs (*Lowy-Gallego et al., 2019*) at a maximum distance of 25 kb from the CTSS with maximum pooled CAGE signal within each promoter for association with changes in promoter expression levels or decomposed promoter contribution. Resulting p-values were FDR-adjusted according to the total number of promoters or decomposed promoters tested genome-wide within the MatrixEQTL R package. prQTLs, dprQTLs, and frQTLs with FDR ≤5% were retained. A promoter was associated with a dprQTL or frQTL if at least one of its decomposed promoters was associated with a dprQTL or frQTL at FDR < 5%.

## Acknowledgements

We thank members of the Andersson lab for rewarding discussions. Sequencing was performed by the SNP&SEQ Technology Platform in Uppsala, part of the National Genomics Infrastructure (NGI) Sweden and Science for Life Laboratory. The SNP&SEQ Platform is also supported by the Swedish Research Council and the Knut and Alice Wallenberg Foundation.

## Additional information

### Funding

| Funder | Grant reference number | Author |
|---|---|---|
| Danmarks Frie Forskningsfond | 6108-00038 | Robin Andersson |
| European Research Council | 638173 | Robin Andersson |
| Novo Nordisk Fonden | NNF18OC0052570 | Robin Andersson |
| Novo Nordisk Fonden | NNF20OC0059796 | Robin Andersson |

The funders had no role in study design, data collection and interpretation, or the decision to submit the work for publication.

### Author contributions

Hjörleifur Einarsson, Conceptualization, Data curation, Software, Formal analysis, Investigation, Visualization, Methodology, Writing – original draft, Writing – review and editing; Marco Salvatore, Software, Formal analysis, Investigation, Methodology, Writing – review and editing; Christian Vaagensø, Investigation, Writing – review and editing, Performed CAGE experiments; Nicolas Alcaraz, Data curation, Software, Methodology, Writing – review and editing; Jette Bornholdt, Investigation, Writing – review and editing, Contributed to CAGE experiments; Sarah Rennie, Software, Methodology, Writing – review and editing; Robin Andersson, Conceptualization, Resources, Software, Formal analysis, Supervision, Funding acquisition, Investigation, Visualization, Methodology, Writing – original draft, Project administration, Writing – review and editing

### Author ORCIDs

Hjörleifur Einarsson (iD) http://orcid.org/0000-0002-0565-9053
Robin Andersson (iD) http://orcid.org/0000-0003-1516-879X

### Decision letter and Author response

Decision letter https://doi.org/10.7554/eLife.80943.sa1
Author response https://doi.org/10.7554/eLife.80943.sa2

## Additional files

### Supplementary files

• Supplementary file 1. LCL sample and CAGE library information. [tab-delimited] Row names: Cell line IDs Columns:

> Sex: The sex of the individual that the cell line was derived from (Male/Female)
> Population: Population origin of the individual the cell line was derived from (YRI/LWK)
> E_Gel: Indicates whether samples underwent a second round of size selection (yes) or not (no)
> CageRun_ID: The CAGE library batch the sample was prepared in, with comma separating IDs if sample was included in multiple runs
> SeqPool_ID: The sequence pool IDs the sample was included in, with comma separating IDs if sample was included in multiple pools
> SeqRun_ID: The sequence run IDs the sample was included in, with comma separating IDs if sample was included in multiple sequencing runs
> Total_reads: Total reads sequenced
> Mapped_reads: Total mapped reads

• Supplementary file 2. CAGE-inferred promoters associated with GENCODE-annotated TSSs. [tab-delimited] Row names: genomic coordinates of promoters provided as chromosome:start-end;strand Columns:

> geneID: Ensembl ID of the associated gene

median: median TPM-normalized tag cluster expression across the LCL panel
CV: coefficient of variation of TPM-normalized tag cluster expression across the LCL panel
mean: mean TPM-normalized tag cluster expression across the LCL panel
log10_CV2: $\log_{10}$-transformed squared coefficient of variation ($CV^2$) of TPM-normalized tag cluster expression across the LCL panel
adjusted_log10_CV2: $\log_{10}$-transformed squared coefficient of variation ($CV^2$) after adjustment of the mean expression-dispersion relationship.
adjusted_quartile: adjusted variability split by quartiles (<Q1: 0–25%, (Q1,Q2]: 25–50%, (Q2,Q3]: 50–75%,>Q3: 75–100%)

• Supplementary file 3. Gene level expression characterization across GM12878 single cells. [tab-delimited] Row names: Ensembl IDs Columns:

median: median scran-normalized read counts across GM12878 single cells
CV: coefficient of variation of scran-normalized read counts across GM12878 single cells
mean: mean scran-normalized read counts across GM12878 single cells
log10_CV2: $\log_{10}$-transformed squared coefficient of variation ($CV^2$) of scran-normalized read counts across GM12878 single cells
adjusted_log10_CV2: $\log_{10}$-transformed squared coefficient of variation ($CV^2$) after adjustment of the mean expression-dispersion relationship across GM12878 single cells

• Supplementary file 4. Promoter differential expression results using DESeq2 in GM12878 after TNFα treatment. [tab-delimited] Row names: genomic coordinates of promoters provided as chromosome:start-end;strand Columns:

baseMean: Average normalized count values
log2FoldChange: Effect size estimate [$\log_2$ Fold Change] for 6 h TNFα/untreated
lfgSE: Standard error estimate for $\log_2$ fold change for 6 h TNFα/untreated
stat: Wald test statistics [Z-statistic]
pvalue: Wald test p-value for 6 h TNFα/untreated
padj: Benjamin-Hochberg corrected p-value
geneID: Ensembl ID of the associated gene
adjusted_log10_CV2: $\log_{10}$-transformed squared coefficient of variation ($CV^2$) after adjustment of the mean expression-dispersion relationship
adjusted_quartile: adjusted variability split by quartiles (<Q1: 0–25%, (Q1,Q2]: 25–50%, (Q2,Q3]: 50–75%,>Q3: 75–100%)

• Supplementary file 5. Decomposed promoter expression characterization and its association with original promoter expression variability. [tab-delimited] Row names: genomic coordinates of decomposed tag clusters provided as chromosome:start-end;strand Columns:

median: median TPM-normalized decomposed tag cluster expression across the LCL panel
CV: coefficient of variation of TPM-normalized decomposed tag cluster expression across the LCL panel
mean: mean TPM-normalized decomposed tag cluster expression across the LCL panel
log10_CV2: $\log_{10}$-transformed squared coefficient of variation ($CV^2$) of TPM-normalized decomposed tag cluster expression across the LCL panel
adjusted_log10_CV2: $\log_{10}$-transformed squared coefficient of variation ($CV^2$) after adjustment of the mean expression-dispersion relationship for decomposed tag cluster..
broad_name: genomic coordinates of the original encompassing promoter provided as chromosome:start-end;strand
broad_adjusted_log10_CV2: $\log_{10}$-transformed squared coefficient of variation ($CV^2$) after adjustment of the mean expression-dispersion relationship for the original encompassing promoter.
broad_adjusted_quartile: adjusted variability of the original encompassing promoter split by quartiles (<Q1: 0–25%, (Q1,Q2]: 25–50%, (Q2,Q3]: 50–75%,>Q3: 75–100%)

• Supplementary file 6. Lead prQTL hit for each promoter. [tab-delimited] Row names: row number Columns:

SNP: lead SNP (lowest FDR) genomic coordinates identified as prQTL for the corresponding promoter.

gene: genomic coordinates of promoter associated with given lead SNP provided as chromosome:start-end;strand

beta: Effect size estimate for the given SNP-promoter pair

t.stat: Test statistic (t-statistic) for the given SNP-promoter pair

p.value: p-value for given SNP-promoter pair

FDR: False discovery rate estimated using Benjamini–Hochberg procedure

• Supplementary file 7. Lead frQTL hit for each promoter. [tab-delimited] Row names: row number Columns:

broadID: genomic coordinates of original promoters overlapping decomposed promoters associated with frQTL provided as chromosome:start-end;strand

decomposedID: genomic coordinates of the decomposed promoters with the strongest association (FDR) with given lead SNP provided as chromosome:start-end;strand.

geneID: Ensembl ID of the associated gene

SNP: lead SNP (lowest FDR and FDR ≤ 5%) genomic coordinates identified as frQTL for the corresponding promoter.

beta: Effect size estimate for the given SNP-decomposed promoter pair

FDR: False discovery rate estimated using Benjamini–Hochberg procedure

relative_effect: relative change in original promoter expression between major and minor allele for given SNP. Calculated as (A-B)/A using mean TPM-normalized expression for A (homozygous for major allele) and B (homozygous for minor allele).

adjusted_quartile: adjusted variability of original promoters split by quartiles (<Q1: 0–25%, (Q1,Q2]: 25–50%, (Q2,Q3]: 50–75%,>Q3: 75–100%)

• Supplementary file 8. Gene level expression characterization across LCLs using RNA-seq (Geuvadis). [tab-delimited] Row names: Ensembl ID of the associated gene Columns:

symbol: gene symbol (HGNC ID)

median: median TPM-normalized gene expression across the LCL panel

CV: coefficient of variation of TPM-normalized gene expression across the LCL panel

mean: mean TPM-normalized gene expression across the LCL panel

log10_CV2: $\log_{10}$-transformed squared coefficient of variation ($CV^2$) of TPM-normalized gene expression across the LCL panel

adjusted_log10_CV2: $\log_{10}$-transformed squared coefficient of variation ($CV^2$) after adjustment of the mean gene expression-dispersion relationship.

• Supplementary file 9. Promoters associated with stabilizing frQTLs. [tab-delimited] Row names: row number Columns:

broadID: genomic coordinates of original promoters overlapping decomposed promoters associated with stabilizing frQTL provided as chromosome:start-end;strand

geneID: Ensembl ID of the associated gene

SNP: Lead SNPs identified as stabilizing frQTLs for given promoter

• MDAR checklist

### Data availability

CAGE data were deposited into the Gene Expression Omnibus (GEO) database under accession number GSE188131 (https://www.ncbi.nlm.nih.gov/geo/query/acc.cgi?acc=GSE188131). GM12878 scRNA-seq data were retrieved from GEO (accession number GSE126321). Processed Geuvadis RNA-seq gene expression data were retrieved from recount2 (*Collado-Torres et al., 2017*) (accession number ERP001942). Processed GTEx RNA-seq gene expression data were retrieved from the GTEx portal (https://www.gtexportal.org/home/datasets, version-8). MNase-seq data were retreived from GEO (accession number GSE36979). Predicted transcription factor binding sites for 746 TFs were obtained from *Fornes et al., 2020* (http://expdata.cmmt.ubc.ca/JASPAR/downloads/UCSC_tracks/2020/hg38/). Code for data analysis performed in this study is publicly available on GitHub:

https://github.com/anderssonlab/Einarsson_et_al_2022/, (copy archived at swh:1:rev:de2e6b8a35c-16687c8d55630e65c78489a629c99; *Andersson, 2022*).

The following dataset was generated:

| Author(s) | Year | Dataset title | Dataset URL | Database and Identifier |
|---|---|---|---|---|
| Einarsson H | 2021 | Promoter sequence and architecture determine expression variability and confer robustness of genetic variation | https://www.ncbi.nlm.nih.gov/geo/query/acc.cgi?acc=GSE188131 | NCBI Gene Expression Omnibus, GSE188131 |

The following previously published datasets were used:

| Author(s) | Year | Dataset title | Dataset URL | Database and Identifier |
|---|---|---|---|---|
| Osorio D, Yu X, Yu P, Serpedin E, Cai JJ | 2019 | Single-cell RNA sequencing of lymphoblastoid cell lines of European and African ancestries | https://www.ncbi.nlm.nih.gov/geo/query/acc.cgi?acc=GSE126321 | NCBI Gene Expression Omnibus, GSE126321 |
| GTEx Consortium | 2017 | GTEx Analysis V8 | https://storage.googleapis.com/gtex_analysis_v8/rna_seq_data/GTEx_Analysis_2017-06-05_v8_RNASeQCv1.1.9_gene_median_tpm.gct.gz | dbGaP, phs000424.v8.p2 |
| Gaffney DJ, Pai AA, Fondufe-Mittendorf YN, Lewellen N, Michelini K, Gilad Y, Pritchard JK | 2012 | Genome-wide maps of nucleosome occupancy in human lymphoblastoid cell lines | https://www.ncbi.nlm.nih.gov/geo/query/acc.cgi?acc=GSE36979 | NCBI Gene Expression Omnibus, GSE36979 |

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

# Appendix 1

**Appendix 1—key resources table**

| Reagent type (species) or resource | Designation | Source or reference | Identifiers | Additional information |
|---|---|---|---|---|
| cell line (*Homo-sapiens*) | Lymphoblastoid cell line | Coriell | GM18505 | |
| cell line (*Homo-sapiens*) | Lymphoblastoid cell line | Coriell | GM18507 | |
| cell line (*Homo-sapiens*) | Lymphoblastoid cell line | Coriell | GM19238 | |
| cell line (*Homo-sapiens*) | Lymphoblastoid cell line | Coriell | GM19239 | |
| cell line (*Homo-sapiens*) | Lymphoblastoid cell line | Coriell | GM18879 | |
| cell line (*Homo-sapiens*) | Lymphoblastoid cell line | Coriell | GM18501 | |
| cell line (*Homo-sapiens*) | Lymphoblastoid cell line | Coriell | GM18876 | |
| cell line (*Homo-sapiens*) | Lymphoblastoid cell line | Coriell | GM18877 | |
| cell line (Homo-sapiens) | Lymphoblastoid cell line | Coriell | GM18878 | |
| cell line (Homo-sapiens) | Lymphoblastoid cell line | Coriell | GM19206 | |
| cell line (Homo-sapiens) | Lymphoblastoid cell line | Coriell | GM19043 | |
| cell line (Homo-sapiens) | Lymphoblastoid cell line | Coriell | GM18487 | |
| cell line (Homo-sapiens) | Lymphoblastoid cell line | Coriell | GM18486 | |
| cell line (Homo-sapiens) | Lymphoblastoid cell line | Coriell | GM19209 | |
| cell line (Homo-sapiens) | Lymphoblastoid cell line | Coriell | GM19153 | |
| cell line (Homo-sapiens) | Lymphoblastoid cell line | Coriell | GM18881 | |
| cell line (Homo-sapiens) | Lymphoblastoid cell line | Coriell | GM18517 | |
| cell line (Homo-sapiens) | Lymphoblastoid cell line | Coriell | GM19144 | |
| cell line (Homo-sapiens) | Lymphoblastoid cell line | Coriell | GM19210 | |
| cell line (Homo-sapiens) | Lymphoblastoid cell line | Coriell | GM18508 | |
| cell line (Homo-sapiens) | Lymphoblastoid cell line | Coriell | GM19099 | |
| cell line (Homo-sapiens) | Lymphoblastoid cell line | Coriell | GM18489 | |
| cell line (Homo-sapiens) | Lymphoblastoid cell line | Coriell | GM19223 | |
| cell line (Homo-sapiens) | Lymphoblastoid cell line | Coriell | GM18853 | |

*Appendix 1 Continued on next page*

*Appendix 1 Continued*

| Reagent type (species) or resource | Designation | Source or reference | Identifiers | Additional information |
|---|---|---|---|---|
| cell line (Homo-sapiens) | Lymphoblastoid cell line | Coriell | GM18916 | |
| cell line (Homo-sapiens) | Lymphoblastoid cell line | Coriell | GM19147 | |
| cell line (Homo-sapiens) | Lymphoblastoid cell line | Coriell | GM19257 | |
| cell line (Homo-sapiens) | Lymphoblastoid cell line | Coriell | GM19131 | |
| cell line (Homo-sapiens) | Lymphoblastoid cell line | Coriell | GM19119 | |
| cell line (Homo-sapiens) | Lymphoblastoid cell line | Coriell | GM19201 | |
| cell line (Homo-sapiens) | Lymphoblastoid cell line | Coriell | GM19204 | |
| cell line (Homo-sapiens) | Lymphoblastoid cell line | Coriell | GM19092 | |
| cell line (Homo-sapiens) | Lymphoblastoid cell line | Coriell | GM19130 | |
| cell line (Homo-sapiens) | Lymphoblastoid cell line | Coriell | GM19137 | |
| cell line (Homo-sapiens) | Lymphoblastoid cell line | Coriell | GM19102 | |
| cell line (Homo-sapiens) | Lymphoblastoid cell line | Coriell | GM19159 | |
| cell line (Homo-sapiens) | Lymphoblastoid cell line | Coriell | GM18871 | |
| cell line (Homo-sapiens) | Lymphoblastoid cell line | Coriell | GM19200 | |
| cell line (Homo-sapiens) | Lymphoblastoid cell line | Coriell | GM19171 | |
| cell line (Homo-sapiens) | Lymphoblastoid cell line | Coriell | GM19207 | |
| cell line (Homo-sapiens) | Lymphoblastoid cell line | Coriell | GM18516 | |
| cell line (Homo-sapiens) | Lymphoblastoid cell line | Coriell | GM18499 | |
| cell line (Homo-sapiens) | Lymphoblastoid cell line | Coriell | GM19143 | |
| cell line (Homo-sapiens) | Lymphoblastoid cell line | Coriell | GM19093 | |
| cell line (Homo-sapiens) | Lymphoblastoid cell line | Coriell | GM19172 | |
| cell line (Homo-sapiens) | Lymphoblastoid cell line | Coriell | GM19098 | |
| cell line (Homo-sapiens) | Lymphoblastoid cell line | Coriell | GM18520 | |
| cell line (Homo-sapiens) | Lymphoblastoid cell line | Coriell | GM19152 | |
| cell line (Homo-sapiens) | Lymphoblastoid cell line | Coriell | GM19116 | |

*Appendix 1 Continued*

| Reagent type (species) or resource | Designation | Source or reference | Identifiers | Additional information |
|---|---|---|---|---|
| cell line (Homo-sapiens) | Lymphoblastoid cell line | Coriell | GM19138 | |
| cell line (Homo-sapiens) | Lymphoblastoid cell line | Coriell | GM18504 | |
| cell line (Homo-sapiens) | Lymphoblastoid cell line | Coriell | GM19036 | |
| cell line (Homo-sapiens) | Lymphoblastoid cell line | Coriell | GM18870 | |
| cell line (Homo-sapiens) | Lymphoblastoid cell line | Coriell | GM19310 | |
| cell line (Homo-sapiens) | Lymphoblastoid cell line | Coriell | GM18511 | |
| cell line (Homo-sapiens) | Lymphoblastoid cell line | Coriell | GM19222 | |
| cell line (Homo-sapiens) | Lymphoblastoid cell line | Coriell | GM19038 | |
| cell line (Homo-sapiens) | Lymphoblastoid cell line | Coriell | GM19046 | |
| cell line (Homo-sapiens) | Lymphoblastoid cell line | Coriell | GM19314 | |
| cell line (Homo-sapiens) | Lymphoblastoid cell line | Coriell | GM19313 | |
| cell line (Homo-sapiens) | Lymphoblastoid cell line | Coriell | GM19044 | |
| cell line (Homo-sapiens) | Lymphoblastoid cell line | Coriell | GM19020 | |
| cell line (Homo-sapiens) | Lymphoblastoid cell line | Coriell | GM18873 | |
| cell line (Homo-sapiens) | Lymphoblastoid cell line | Coriell | GM18907 | |
| cell line (Homo-sapiens) | Lymphoblastoid cell line | Coriell | GM18909 | |
| cell line (Homo-sapiens) | Lymphoblastoid cell line | Coriell | GM18868 | |
| cell line (Homo-sapiens) | Lymphoblastoid cell line | Coriell | GM18910 | |
| cell line (Homo-sapiens) | Lymphoblastoid cell line | Coriell | GM18908 | |
| cell line (Homo-sapiens) | Lymphoblastoid cell line | Coriell | GM19095 | |
| cell line (Homo-sapiens) | Lymphoblastoid cell line | Coriell | GM19107 | |
| cell line (Homo-sapiens) | Lymphoblastoid cell line | Coriell | GM18867 | |
| cell line (*Homo-sapiens*) | Lymphoblastoid cell line | Coriell | GM19108 | |
| cell line (*Homo-sapiens*) | Lymphoblastoid cell line | Coriell | GM19121 | |
| cell line (*Homo-sapiens*) | Lymphoblastoid cell line | Coriell | GM19117 | |

*Appendix 1 Continued*

| Reagent type (species) or resource | Designation | Source or reference | Identifiers | Additional information |
| --- | --- | --- | --- | --- |
| cell line (*Homo-sapiens*) | Lymphoblastoid cell line | Coriell | GM19175 | |
| cell line (*Homo-sapiens*) | Lymphoblastoid cell line | Coriell | GM19184 | |
| cell line (*Homo-sapiens*) | Lymphoblastoid cell line | Coriell | GM19213 | |
| cell line (*Homo-sapiens*) | Lymphoblastoid cell line | Coriell | GM18519 | |
| cell line (*Homo-sapiens*) | Lymphoblastoid cell line | Coriell | GM18502 | |
| cell line (Homo-sapiens) | Lymphoblastoid cell line | Coriell | GM19113 | |
| cell line (Homo-sapiens) | Lymphoblastoid cell line | Coriell | GM19028 | |
| cell line (Homo-sapiens) | Lymphoblastoid cell line | Coriell | GM19041 | |
| cell line (Homo-sapiens) | Lymphoblastoid cell line | Coriell | GM19307 | |
| cell line (Homo-sapiens) | Lymphoblastoid cell line | Coriell | GM19031 | |
| cell line (Homo-sapiens) | Lymphoblastoid cell line | Coriell | GM18874 | |
| cell line (Homo-sapiens) | Lymphoblastoid cell line | Coriell | GM19118 | |
| cell line (Homo-sapiens) | Lymphoblastoid cell line | Coriell | GM19190 | |
| cell line (Homo-sapiens) | Lymphoblastoid cell line | Coriell | GM19149 | |
| cell line (Homo-sapiens) | Lymphoblastoid cell line | Coriell | GM19248 | |
| cell line (Homo-sapiens) | Lymphoblastoid cell line | Coriell | GM18934 | |
| cell line (Homo-sapiens) | Lymphoblastoid cell line | Coriell | GM19114 | |
| cell line (Homo-sapiens) | Lymphoblastoid cell line | Coriell | GM19146 | |
| cell line (Homo-sapiens) | Lymphoblastoid cell line | Coriell | GM18923 | |
| cell line (Homo-sapiens) | Lymphoblastoid cell line | Coriell | GM18924 | |
| cell line (Homo-sapiens) | Lymphoblastoid cell line | Coriell | GM18933 | |
| cell line (Homo-sapiens) | Lymphoblastoid cell line | Coriell | GM18917 | |
| cell line (Homo-sapiens) | Lymphoblastoid cell line | Coriell | GM19214 | |
| cell line (Homo-sapiens) | Lymphoblastoid cell line | Coriell | GM19185 | |
| cell line (Homo-sapiens) | Lymphoblastoid cell line | Coriell | GM19027 | |

*Appendix 1 Continued on next page*

*Appendix 1 Continued*

| Reagent type (species) or resource | Designation | Source or reference | Identifiers | Additional information |
|---|---|---|---|---|
| cell line (Homo-sapiens) | Lymphoblastoid cell line | Coriell | GM19225 | |
| cell line (Homo-sapiens) | Lymphoblastoid cell line | Coriell | GM19198 | |
| cell line (Homo-sapiens) | Lymphoblastoid cell line | Coriell | GM19035 | |
| cell line (Homo-sapiens) | Lymphoblastoid cell line | Coriell | GM19197 | |
| cell line (Homo-sapiens) | Lymphoblastoid cell line | Coriell | GM19235 | |
| cell line (Homo-sapiens) | Lymphoblastoid cell line | Coriell | GM18858 | |
| cell line (Homo-sapiens) | Lymphoblastoid cell line | Coriell | GM19026 | |
| cell line (Homo-sapiens) | Lymphoblastoid cell line | Coriell | GM18865 | |
| cell line (Homo-sapiens) | Lymphoblastoid cell line | Coriell | GM19025 | |
| cell line (Homo-sapiens) | Lymphoblastoid cell line | Coriell | GM18915 | |
| cell line (Homo-sapiens) | Lymphoblastoid cell line | Coriell | GM19030 | |
| cell line (Homo-sapiens) | Lymphoblastoid cell line | Coriell | GM19037 | |
| cell line (Homo-sapiens) | Lymphoblastoid cell line | Coriell | GM19024 | |
| cell line (Homo-sapiens) | Lymphoblastoid cell line | Coriell | GM19019 | |
| cell line (Homo-sapiens) | Lymphoblastoid cell line | Coriell | GM18864 | |
| cell line (Homo-sapiens) | Lymphoblastoid cell line | Coriell | GM18523 | |
| cell line (Homo-sapiens) | Lymphoblastoid cell line | Coriell | GM19017 | |
| cell line (Homo-sapiens) | Lymphoblastoid cell line | Coriell | GM18522 | |
| cell line (Homo-sapiens) | Lymphoblastoid cell line | Coriell | GM18488 | |
| cell line (Homo-sapiens) | Lymphoblastoid cell line | Coriell | GM19247 | |
| cell line (Homo-sapiens) | Lymphoblastoid cell line | Coriell | GM18510 | |
| cell line (Homo-sapiens) | Lymphoblastoid cell line | Coriell | GM18856 | |
| cell line (Homo-sapiens) | Lymphoblastoid cell line | Coriell | GM18912 | |
| cell line (Homo-sapiens) | Lymphoblastoid cell line | Coriell | GM18861 | |
| cell line (Homo-sapiens) | Lymphoblastoid cell line | Coriell | GM19141 | |

*Appendix 1 Continued*

| Reagent type (species) or resource | Designation | Source or reference | Identifiers | Additional information |
|---|---|---|---|---|
| cell line (Homo-sapiens) | Lymphoblastoid cell line | Coriell | GM19160 | |
| cell line (Homo-sapiens) | Lymphoblastoid cell line | Coriell | GM12878 | |
| sequence-based reagent | MPG beads, 10 ml | Pure biotech | MSTR0510 | |
| sequence-based reagent | AMPure, 60 ml | Ramcon | A63881 | |
| sequence-based reagent | RNAClean, 40 ml | Ramcon | A63987 | |
| sequence-based reagent | Phusion | Th.Geyer, Finnzymes | M0530L | |
| sequence-based reagent | PrimeScript | TaKaRa | 2680 A | |
| sequence-based reagent | RNAseONE | Th.Geyer | M4265 | |
| sequence-based reagent | ddH2O | VWR | | |
| commercial assay or kit | MinElute PCR purification kit (250 columns) | Qiagen | #28006 | |
| sequence-based reagent | LA Taq | TaKaRa | RR002A | |
| sequence-based reagent | DNA1000 kit | Agilent | 5067–1504 | |
| sequence-based reagent | RNA pico kit | Agilent | 5067–1513 | |
| sequence-based reagent | Biotin (long arm) hydrazide, 50 mg | VWR | Vectsp-1100 | |
| sequence-based reagent | E-gel sizeselect | Lifetech | G6610-02 | |
| commercial assay or kit | PureLink Dnase set | Lifetech | #12185010 | |
| sequence-based reagent | EcoP15I, 2500 U | Th.Geyer, NEB | N/R0646L | |
| sequence-based reagent | Sinefungin, 2 mg | Merck, Calbiochem-Novabiochem International | #567051 | |
| sequence-based reagent | Antarctic phosphatase, 5000 U | Bionordika, NEB | M0289L | |
| sequence-based reagent | Trehalose dihydrat | Sigma | Y0001172-1EA | |
| sequence-based reagent | d-Sorbitol | VWR | 85529–250 G | |
| sequence-based reagent | NaIO4, 5 g | Sigma | 311448–5 G | |
| sequence-based reagent | *E. coli* tRNA, 500 U | Sigma | R1753-500UN | ribonucleic acid, transfer from *Escherichia coli* Type XX, Strain W, lyophilized powder |
| sequence-based reagent | RQ1 RNase-free DNase | Promega | M6101 | |

*Appendix 1 Continued on next page*

*Appendix 1 Continued*

| Reagent type (species) or resource | Designation | Source or reference | Identifiers | Additional information |
|---|---|---|---|---|
| sequence-based reagent | Proteinase K | Lifetech | 25530–049 | |
| sequence-based reagent | ATP, 10 mM | Bionordika, NEB | P0756S | |
| sequence-based reagent | Trizol LS, 100 ml | Lifetech, Invitrogen | 10296–010 | |
| sequence-based reagent | DNA ligation kit, Mighty Mix | TaKaRa | 00006023 TAKARA | |
| sequence-based reagent | T4 DNA ligase, 20000 U | Th.Geyer, NEB | N/M0202L | |
| sequence-based reagent | Exonuclease I, 3.000 U | Th.Geyer | N/M0293S | |
| sequence-based reagent | dNTPs, 10 mM, 1 ml | Kælder | | |
| sequence-based reagent | Sodium acetate | Sigma | S2889-250G | |
| sequence-based reagent | Sodium citrate, 500 g | Sigma, MP Biomedicals | W302600-1KG-K | |
| sequence-based reagent | EDTA (4x100 ml, 0.5 M pH = 8.0) | Lifetech | 15575–020 | |
| sequence-based reagent | PCR SYBR mix 2*5 mL | Lifetech | 4364344 | |
| other | Penicillin-Streptomycin | ThermoFisher Scientific | 15140122 | Cell culture supplement |
| other | L-Glutamine | ThermoFisher Scientific | A2916801 | Cell culture supplement |
| other | RPMI 1640 | ThermoFisher Scientific | 11875083 | Cell culture media |
| other | Fetal bovine serum | ThermoFisher Scientific | A3840302 | Cell culture supplement |
| peptide, recombinant protein | TNFa | R&DSystems | P01375 | 25 ng/ml |

