## [Editor Report]

This paper presents valuable findings about how human genetic variation impacts gene expression. Using a compelling analysis of new experimental data based on cell lines from 108 individuals, the authors uncover features that distinguish promoters with highly variable expression across individuals from those exhibiting low variability. This work and the associated resource will be of broad interest for further investigations of the interplay between genetic variation and gene expression control.

---

## [Decision Letter]

**Decision letter after peer review:**

Thank you for submitting your article "Promoter sequence and architecture determine expression variability and confer robustness to genetic variants" for consideration by *eLife*. Your article has been reviewed by 2 peer reviewers, and the evaluation has been overseen by a Reviewing Editor and George Perry as the Senior Editor. The reviewers have opted to remain anonymous.

Essential revisions:

1) Please consider alternative interpretations for the observations as suggested by the reviewers, and perform additional checks to consider which ones may be more plausible.

2) It would be great if the authors could provide additional support for promoter plasticity, or provide an alternative explanation that is better supported by their analyses.

3) The reviewers have made a case to address the role of initiator (YR) motifs and nucleosome positioning, especially regarding the decomposed promoters.

4) Additionally, the reviewers suggested considering how the possible heterogeneity of gene expression may have an impact on the interpretation of the results.

5) Please reassess the interpretations of observed correlations as causation. A general change to a more cautious language that follows more closely the data is overemphasised is advised.

6) Improved clarity on some of the descriptions is also requested, like the one mentioned by the reviewers about the ETS family.

*Reviewer #1 (Recommendations for the authors):*

– I think that the data is great and that you analysed it well. I just think that many of your interpretations of the results are not the only possible ones, and at least some are unlikely. I have addressed this in the public part of the review and would recommend that you consider each point and perform a couple of additional checks where they can decide which interpretation is more likely.

– The role of initiator (YR) motifs and nucleosome positioning, especially with regard to the decomposed promoters, should be addressed.

– Anecdotally, from looking at many GTEx expression profiles of individual genes, genes in EBV-immortalised lymphoblastoid cell lines often show expression levels of multiple otherwise broadly expressed genes that are outliers to all other cell/tissue types in the corresponding panels. How much is known about heterogeneity of gene expression across EBV-transformed cells in general, and how would that affect the interpretation of your analyses?

*Reviewer #2 (Recommendations for the authors):*

– The authors use language that pimples a causal relationship. A great example is on the bottom of page 10: "promoter sequence can influence both low and high" (the word "influence"); and "several TFs were identified as contributing partially to the variability in promoter expression" (the word "contributing"). Another example is the transition into the next section, where the authors state that "[DNA sequence motifs provide] distinct mechanisms for attenuating or amplifying variability".

The authors do not directly show that these motifs are casual for promoter variability, and I'm not inclined to believe they are causal. I think the most natural model is one where natural selection constrains the expression of genes with some promoter architectures (i.e., housekeeping, CpG island, ubiquitous expressed) more than others because of their important biological function. I think the authors should use language indicative of correlation rather than causation throughout these sections.

– The authors state on page 10 that "individual ETS family members are independently strong predictors of low promoter variability". But I think the ETS family members being identified simply reflects their shared DNA sequence, as the authors imply in the next sentence. I think it's confusing to readers to state this as a result, then state the obvious technical reason it happens. I think this section should be reworked.

– Much of the evidence for genetic plasticity currently involves correlations between promoter variability (defined as the squared coefficient of variation across the population) and either piQTL presence or effect size. I do not find these correlations to be all that convincing to the argument for plasticity as it stands. Promoter variability, as defined by the authors, essentially measures variation across the population, which will also lead to that promoter being more likely to have a piQTL. Thus, the logic seems at least somewhat circular as it stands. By contrast, I found the arguments around frQTLs to be much more convincing – the idea that specific SNPs can change the fraction of a decomposed promoter's contribution is much more helpful to the authors' argument. As it stands, though, this is a fairly minor portion of the manuscript buried in the results paragraphs. I think it would help if the authors played this up a bit more.

Along these same lines, I wonder if the authors have individually tested the decomposed promoters for decomposed-prQTLs (maybe dprQTLs)? Identifying nearby dpr-QTLs that have opposite effects could also support the author's argument in a way that does not follow by definition.

---

## [Author Response]

Reviewer #1 (Recommendations for the authors):– I think that the data is great and that you analysed it well. I just think that many of your interpretations of the results are not the only possible ones, and at least some are unlikely. I have addressed this in the public part of the review and would recommend that you consider each point and perform a couple of additional checks where they can decide which interpretation is more likely.

We thank the reviewer for constructive criticism and for good suggestions on how to improve the manuscript. In our revised manuscript we have followed the recommendations by the reviewer and:

– Toned down implied causal relationships and added additional interpretations to our results, including YR positional preferences and the suggested model of competition between core promoters (decomposed promoters) underlying low variable promoters

– Performed additional analyses on nucleosome positioning within decomposed promoters of low variable promoters

In all, we believe these revisions have substantially improved our manuscript and even strengthened our previous conclusions.

– Anecdotally, from looking at many GTEx expression profiles of individual genes, genes in EBV-immortalised lymphoblastoid cell lines often show expression levels of multiple otherwise broadly expressed genes that are outliers to all other cell/tissue types in the corresponding panels. How much is known about heterogeneity of gene expression across EBV-transformed cells in general, and how would that affect the interpretation of your analyses?

Thank you, this is a very good point. Such heterogeneity does not only affect our study but most studies relying on individually generated cell cultures. Previous studies have shown that EBV-immortalized LCLs can in fact show genotype-independent variability, due to factors not fully understood. In the revised version of the manuscript, we acknowledge that using immortalized cell cultures has its caveates when studying expression variation.

“It is important to note that the regulatory programs of EBV-immortalized LCLs, like other cultured cells, have been shown to be susceptible to genotype-independent sources of variation, such as primary cellular heterogeneity and EBV’s viral cellular reprogramming (Choy et al., 2008, DOI: 10.1371/journal.pgen.1000287; Ozgyin et al., 2019, DOI: 10.1038/s41598-019-40897-9). While we cannot exclude that such sources may influence the measured variability of some genes, we speculate that non-genetic variation would rather dampen than elevate the associations and effect sizes we report here, in particular the decomposed promoter switches highlighted by our genotype association analysis.”

Hence, one could speculate that the phenomenon of stabilizing promoter switches is even more common than what we can observe with our current data.

Reviewer #2 (Recommendations for the authors):– The authors use language that pimples a causal relationship. A great example is on the bottom of page 10: "promoter sequence can influence both low and high" (the word "influence"); and "several TFs were identified as contributing partially to the variability in promoter expression" (the word "contributing"). Another example is the transition into the next section, where the authors state that "[DNA sequence motifs provide] distinct mechanisms for attenuating or amplifying variability".The authors do not directly show that these motifs are casual for promoter variability, and I'm not inclined to believe they are causal. I think the most natural model is one where natural selection constrains the expression of genes with some promoter architectures (i.e., housekeeping, CpG island, ubiquitous expressed) more than others because of their important biological function. I think the authors should use language indicative of correlation rather than causation throughout these sections.– The authors state on page 10 that "individual ETS family members are independently strong predictors of low promoter variability". But I think the ETS family members being identified simply reflects their shared DNA sequence, as the authors imply in the next sentence. I think it's confusing to readers to state this as a result, then state the obvious technical reason it happens. I think this section should be reworked.

We thank the reviewer for pointing out these concerns. We have carefully reworded the mentioned sentences and other conclusions of the text to better reflect our findings and to clarify their meanings.

– Much of the evidence for genetic plasticity currently involves correlations between promoter variability (defined as the squared coefficient of variation across the population) and either piQTL presence or effect size. I do not find these correlations to be all that convincing to the argument for plasticity as it stands. Promoter variability, as defined by the authors, essentially measures variation across the population, which will also lead to that promoter being more likely to have a piQTL. Thus, the logic seems at least somewhat circular as it stands. By contrast, I found the arguments around frQTLs to be much more convincing – the idea that specific SNPs can change the fraction of a decomposed promoter's contribution is much more helpful to the authors' argument. As it stands, though, this is a fairly minor portion of the manuscript buried in the results paragraphs. I think it would help if the authors played this up a bit more.

We thank the reviewer for this comment. We do not dispute that more variable promoters are more likely to be associated with a prQTL. However, we feel that showing that this is in fact the case further highlights the interesting finding of frQTLs of low variable promoters. Despite the fact that these promoters do not show strong prQTL effect sizes, we see evidence of variability in TSS usage, as revealed by frQTLs and decomposed promoter eQTLs (dprQTLs, see below). This indicates that low variable promoters are indeed affected by genetic variation but that these effects are masked due to compensatory switches between core promoters.

Along these same lines, I wonder if the authors have individually tested the decomposed promoters for decomposed-prQTLs (maybe dprQTLs)? Identifying nearby dpr-QTLs that have opposite effects could also support the author's argument in a way that does not follow by definition.

We thank the reviewer for this great suggestion. In the revised version of our manuscript, we also tested the association between the genotypes of common genetic variants (MAF ≥ 10%) and the expression of decomposed promoters (decomposed promoter eQTL, dprQTL). Interestingly, while low variable promoters are less likely to have prQTLs with large regulatory effects, dprQTLs exhibited similar prevalence (Supplementary Figure 10C) and maximum effect sizes (Supplementary Figure 10D) across promoter variability classes. We believe these results strengthen our previous results from frQTLs.